# miR-130a and miR-145 reprogram Gr-1$^+$CD11b$^+$ myeloid cells and inhibit tumor metastasis through improved host immunity

Hiroki Ishii [1], Suman K. Vodnala[1], Bhagelu R. Achyut[1,4], Jae Young So[1], M. Christine Hollander[1],
Tim F. Greten [2], Ashish Lal[3] & Li Yang[1]

Tumor-derived soluble factors promote the production of Gr-1$^+$CD11b$^+$ immature myeloid cells, and TGFβ signaling is critical in their immune suppressive function. Here, we report that miR-130a and miR-145 directly target TGFβ receptor II (TβRII) and are down-regulated in these myeloid cells, leading to increased TβRII. Ectopic expression of miR-130a and miR-145 in the myeloid cells decreased tumor metastasis. This is mediated through a downregulation of type 2 cytokines in myeloid cells and an increase in IFNγ-producing cytotoxic CD8 T lymphocytes. miR-130a- and miR-145-targeted molecular networks including TGFβ and IGF1R pathways were correlated with higher tumor stages in cancer patients. Lastly, miR-130a and miR-145 mimics, as well as IGF1R inhibitor NT157 improved anti-tumor immunity and inhibited metastasis in preclinical mouse models. These results demonstrated that miR-130a and miR-145 can reprogram tumor-associated myeloid cells by altering the cytokine milieu and metastatic microenvironment, thus enhancing host antitumor immunity.

[1] Laboratory of Cancer Biology and Genetics, Center for Cancer Research, National Cancer Institute, NIH, Bethesda, MD 20892, USA. [2] Gastrointestinal Malignancy Section, Center for Cancer Research, National Cancer Institute, NIH, Bethesda, MD 20892, USA. [3] Genetic Branch, Center for Cancer Research, National Cancer Institute, NIH, Bethesda, MD 20892, USA. [4] Present address: Tumor Angiogenesis Laboratory, Georgia Cancer Center, Augusta University, Augusta 30912, USA. Correspondence and requests for materials should be addressed to L.Y. (email: yangl3@mail.nih.gov)

Tumor-associated myeloid cells promote distant organ metastasis in hosts bearing solid tumors and are considered a bonafide target for cancer therapy[1,2]. These myeloid cells, including Gr-1$^+$CD11b$^+$ immature myeloid cells or myeloid-derived suppressor cells (MDSCs)[3], tumor-associated macrophages (TAMs)[4] and neutrophils (TANs)[5,6], are intricately connected. Altogether they influence tumor and host micro/macro environment and immune responses. Growth factors, cytokines, chemokines, and inflammatory mediators produced by tumor cells and other regulatory immune cells such as B and regulatory T (Treg) cells facilitate the polarization of myeloid cell function into a type 2 but not type 1 phenotypes, similar to the M1/M2 paradigm for TAMs[7,8]. Transforming growth factor β (TGFβ), interleukin (IL)-10, IL-4, and IL-13 induce type 2 polarization of TAM, which inhibits cytotoxic CD8 T lymphocyte activity thus compromising host anti-tumor immunity[9].

We and others previously reported that myeloid-specific TGFβ signaling is critical in tumor metastasis. Specific deletion of *Tgfbr2*, the gene encoding TGFβ receptor 2 (TβRII), in myeloid cells significantly inhibited tumor metastasis through down-regulation of immunosuppressive cytokines and chemokines[10–12]. Indeed increased TβRII expression and elevated TGFβ signaling pathways have been found to be important in the immune suppressive function of Gr-1$^+$CD11b$^+$ myeloid cells and in progression and metastasis of breast cancer[10,13,14], skin cancer[15], colon cancer[16], and glioma[17]. These studies suggest the importance of myeloid TβRII in metastatic progression and as a potential target for therapy. However, the mechanisms responsible for up-regulating TβRII expression in the myeloid compartment of tumor-bearing hosts remain to be investigated.

MicroRNAs (miRNAs) are an abundant class of small non-coding RNAs and are powerful regulators based on partial sequence complementarity. miRNAs induce mRNA degradation and/or inhibition of protein translation in physiological or pathological conditions such as inflammation and cancer[18–20]. Both oncogenic and tumor suppressive miRNAs have been reported, and they mediate diverse biological functions through targeting multiple mediators in signaling pathways[19,21]. miRNAs are critical in regulating tumor metastasis and microenvironment[22–25], myeloid cell properties and functions[26–29], as well as T cell immune responses[22,30]. Therapeutically, restoration of miRNAs in tumor cells has shown potential for cancer treatment[28,31–34]. Here, we report that miR-130a and miR-145 are down-regulated in myeloid cells from tumor-bearing mice leading to increased TβRII, as well as multiple mediators in IGF1R signaling pathways. Molecules targeted by miR-130a and miR-145 correlated with stages of tumor progression in cancer patients. Further, ectopic expression of miR-130a and miR-145 reprogrammed Gr-1$^+$CD11b$^+$ myeloid cells, skewing a pro-tumor to an anti-tumor microenvironment, and decreased tumor metastasis in several mouse models. Importantly, miR-130a and miR-145 mimics, as well as IGF1R inhibitor NT157 significantly decreased tumor metastasis. Our studies suggest that miR-130a and miR-145-mediated reprogramming of myeloid cells could be a novel approach to target the metastatic microenvironment and improve host anti-tumor immunity.

## Results

### Reduced miR-130a & miR-145 and increased TβRII in Gr-1$^+$CD11b$^+$ cells.

We previously published that myeloid-specific TβRII is significantly elevated, and it plays critical roles in immune suppression and premetastatic niche formation[10,11]. In the current studies, we investigated how TβRII expression is regulated in the myeloid compartment. Gr-1$^+$CD11b$^+$ myeloid cells from 4T1 mammary tumor-bearing mice, which constitute most myeloid cells, have an increased level of TβRII mRNA and

protein when compared to those from healthy control mice (Fig. 1a). The stability of TβRII mRNA in Gr-1$^+$CD11b$^+$ cells from tumor-bearing mice was much greater compared to those from healthy control mice (Fig. 1b) when the myeloid cells were treated with Actinomycin D, a polypeptide antibiotic that interferes with new mRNA synthesis. These results indicate a post-transcription regulation of TβRII.

miRNAs have been shown to regulate myeloid cell function[26–28], and host immune response[22]. We thus examined the differential expression of miRNA in Gr-1$^+$CD11b$^+$ cells from spleens of 4T1 tumor-bearing mice compared to those from healthy control mice (Fig. 1c) using mouse miRNA microarrays (NanoString Technologies), a robust and sensitive method for digital expression detection of over 600 murine miRNAs. Among the differentially expressed miRNAs, miR-130a and miR-145 were down-regulated in Gr-1$^+$CD11b$^+$ myeloid cells from tumor-bearing mice and predicted to target the 3′-untranslated region (UTR) of TβRII mRNA (Fig. 1d; Supplementary Fig. 1a, and Supplementary Table 1). The reduced expression of miR-130a and miR-145 was validated using TaqMan quantitative polymerase chain reaction (qRT-PCR) assays, which was in contrast with miR-19a and miR-93 that were up-regulated (Fig. 1e). miR-130a and miR-145 targeting the TβRII 3′-UTR were further validated using a luciferase reporter assay in which TβRII 3′-UTR was cloned into pGL3 control vector downstream of firefly luciferase. Co-transfection of pGL3 reporter plasmid with miR-130a or miR-145 mimic showed ~40% and 50% reduction in luciferase activity, compared to miR-16 control (Fig. 1f). In a time course experiment in which the Gr-1$^+$CD11b$^+$ cells were sorted from spleens of mice on day 7, 14, 21, or 28 after 4T1 tumor injection in the mammary fat pad (MFP), down-regulation of miR-130 and miR-145 was inversely correlated with the increased levels of TβRII (Fig. 1g). Notably, in both monocytic and granulocytic myeloid subsets, lower miR-130a or miR-145, and higher TβRII expression was observed comparing tumor condition vs healthy condition (Fig. 1h). These results were also observed in a second orthotopic model of E0771 mammary tumors in a C57BL/6 genetic background (Supplementary Fig. 1b, c). Together these data suggest that decreased miR-130a and miR-145 levels are likely responsible for increased TβRII expression in immature myeloid cells.

### Myeloid overexpression of miR-130a and miR-145 reduced metastases.

To test whether miR-130a and miR-145 could decrease TβRII expression in myeloid cells and inhibit tumor metastasis in vivo, miR-130a and miR-145 were subcloned into a pFUGW lentiviral expression vector under control of the CD11b promoter, with or without a GFP reporter (Fig. 2a; upper panel). Lentivirus was used to transduce bone marrow-derived hematopoietic stem/progenitor cells (HS/PCs) (Supplementary Fig. 2a), which were then differentiated into myeloid cells in ex vivo culture with GM-CSF/IL-6 as simulated healthy condition. The Gr-1$^+$CD11b$^+$ myeloid cells were sorted into high, low and negative GFP subsets (Supplementary Fig. 2b). Genomic integration of the miR-130a expression vector in all subsets was detected (Supplementary Fig. 2c). However, the GFP-high and low Gr-1$^+$CD11b$^+$ cells showed ~4 fold more miR-130a expression compared to the GFP-negative cells (Supplementary Fig. 2d; left). TβRII expression was lower in the GFP-high and low Gr-1$^+$CD11b$^+$ cells than that in the GFP-negative Gr-1$^+$CD11b$^+$ cells (Supplementary Fig. 2d; right). These data indicate that vectors were integrated into the genome and were expressed in myeloid cells. Moreover, in ex vivo culture with 4T1 tumor supernatant, these miRNA-engineered HS/PCs were differentiated into CD11b$^+$ myeloid cells (Fig. 2a), and GFP expression was observed in ~68% (Fig. 2b). Within the GFP$^+$CD11b$^+$ population, most were the CD11b$^+$Ly6G$^+$ granulocytic myeloid subset (Fig. 2b). miR-130a and miR-145 were highly

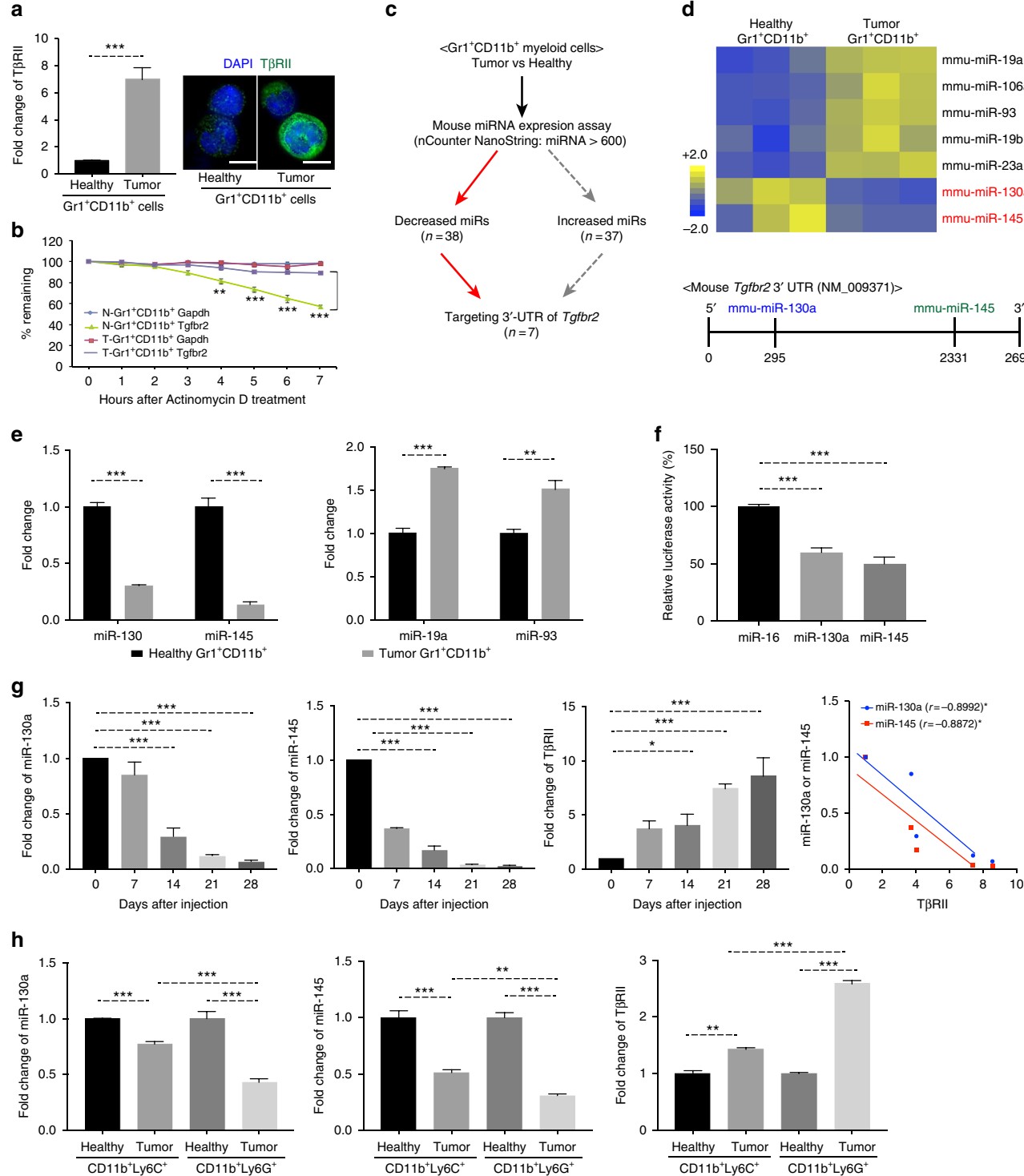

expressed in GFP+ compared to GFP− myeloid cells (Fig. 2c), consistent with a decreased TβRII expression in GFP+ cells (Fig. 2c). These data suggest that miR-130a or miR-145 was induced in myeloid cells by lentiviral expression vectors, and inhibited TβRII expression in myeloid cells.

We then transplanted miRNA-engineered HS/PCs into lethally irradiated recipient mice, which showed good reconstitution 10 weeks after bone marrow transplantation (Supplementary Fig. 2e). Genomic integration of miRNAs was verified in Gr-1+CD11b+ myeloid cells from bone marrow-transplanted mice (Supplementary Fig. 2f). In these experiments, shRNA-TβRII was

used as a positive control as it has better target specificity compared to miRNA approaches. There was no difference in lymphocyte, monocyte, and neutrophil counts comparing miRNA- or Tgfbr2-engineered with the vector control mice that received bone marrow transplantation (Supplementary Fig. 2g). GFP was preferentially expressed in the CD11b+ myeloid lineages but not in CD4+ or CD8+ T and CD19+ B cells of the recipient mice (Supplementary Fig. 2h).

The 4T1 mammary tumor model was used to investigate whether miR-130a and miR-145 repress lung metastasis via down-regulation of TβRII in myeloid cells. This model shares

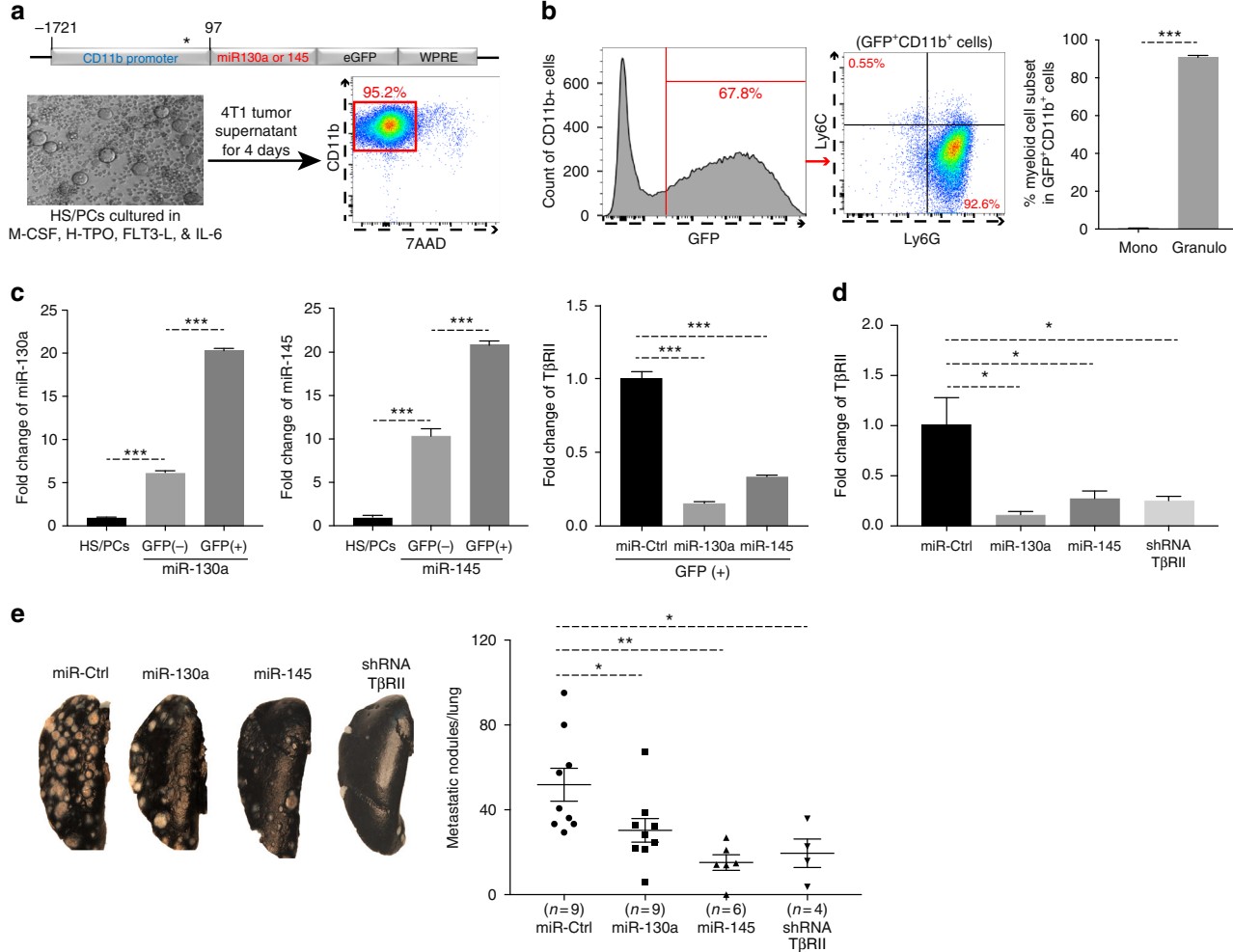

**Fig. 2** Ectopic expression of miR-130a & miR-145 reduced tumor metastases. **a** Top panel: the construct of miR-130a or miR-145 expressions under control of a CD11b promoter, with GFP and WPRE (Woodchuck hepatitis virus posttranscriptional regulatory element) in 3′-UTR. * indicates transcription starting site (TSS); numbers indicate CD11b promoter sequence relative to TSS. Lower panels: microscopy (left), flow cytometry analysis of CD11b+ myeloid cells differentiated from HS/PCs with the lentiviral expression vector in the culture with 4T1 tumor supernatant (right). **b** flow cytometry of GFP expression in CD11b+ myeloid cells (left), and CD11b+Ly6C+ monocytic and CD11b+ Ly6G+ granulocytic subsets in GFP+ myeloid cells (middle) with quantitative data (right). **c** Fold changes of miR-130a (left), and miR-145 (middle) as well as TβRII (right) in sorted GFP+ and GFP- myeloid cells by qRT-PCR, with HS/PCs as a negative control. Data in (**b-c**) are from ex vivo culture as illustrated in (**a**). **d** TβRII fold changes by qRT-PCR from sorted peripheral blood Gr-1+CD11b+ myeloid cells of tumor-bearing mice that received miR-130a, miR-145, and TβRII shRNA-engineered bone marrow transplantation. **e** Lung metastasis of 4T1 cells in mice that received miR-engineered bone marrow transplantation as noted above. 4T1 cells were injected in MFP number 2 and metastatic nodules in the lung were assessed 28 days later ($n = 5$–13 mice). The data are represented as mean±SEM, and Student's $t$ test was performed. *$p < 0.05$, **$p < 0.01$, ***$p < 0.001$

**Fig. 1** miR-130a & miR-145, and TβRII mRNA in Gr-1+CD11b+ cells. **a** Left panels: qRT-PCR for fold changes of TβRII transcript in Gr-1+CD11b+ cells isolated from the spleen of 4T1 tumor-bearing mice (Tumor) compared to that from healthy control mice (Healthy) ($n = 3$). Right panel: Immunofluorescence images of TβRII (green) and DAPI (blue) in Gr-1+CD11b+ cells. scale bar=10 μm. **b** Stability of TβRII and GAPDH transcripts in Gr-1+CD11b+ cells derived from the spleen of 4T1 tumor-bearing mice (T) compared to those from healthy control mice (N). Total RNA was isolated at the indicated times after Actinomycin D treatment. 18sRNA was used to normalize TβRII and GAPDH expression. The experiment was performed three times with three biological replicates in each group. **c** Strategies to identify miRNAs targeting TβRII. **d** Heat map of differentially expressed miRNAs in Gr-1+CD11b+ cells from 4T1 tumor-bearing mice (Tumor) compared to the healthy control (Healthy) ($n = 3$). Yellow and blue colors indicate increased and decreased expression, respectively. Below: Schematic diagram showing the location of miR-130a and miR-145 consensus binding sites in 3′-UTR of TβRII transcript. **e** Expression of the indicated miRNAs by qRT-PCR. qRT-PCR of miR-130a and miR-145 in Gr-1+CD11b+ myeloid cells sorted from the spleen of healthy control or tumor-bearing mice, with miR-19a and miR-93 as controls. **f** Relative luciferase activity of HeLa cells transfected with pGL3-TβRII-3′-UTR vector and miR-16, miR-130a, or miR-145 mimics by electroporation. The miR-16 was used as a negative control. **g** Time course experiments for the expression of miR-130a (left), miR-145 (middle) and TβRII (right) in sorted Gr-1+CD11b+ myeloid cells as measured by qRT-PCR. Y-axis: fold changes; X-axis: days after tumor injection in the MFP ($n = 3$/each time point). Right panel: Spearman analysis of the correlations of miR-130a or miR-145 level with TβRII level. $r$: Pearson's correlation coefficient. **h** qRT-PCR for expression of miR-130a (left), miR-145 (middle) and TβRII (right) in myeloid subsets CD11b+Ly6C+ and CD11b+Ly6G+ sorted from spleens of 4T1 tumor-bearing (Tumor) or healthy control mice (Healthy). The data are represented as mean±SEM, and Student's $t$ test was performed. *$p < 0.05$, **$p < 0.01$, ***$p < 0.001$

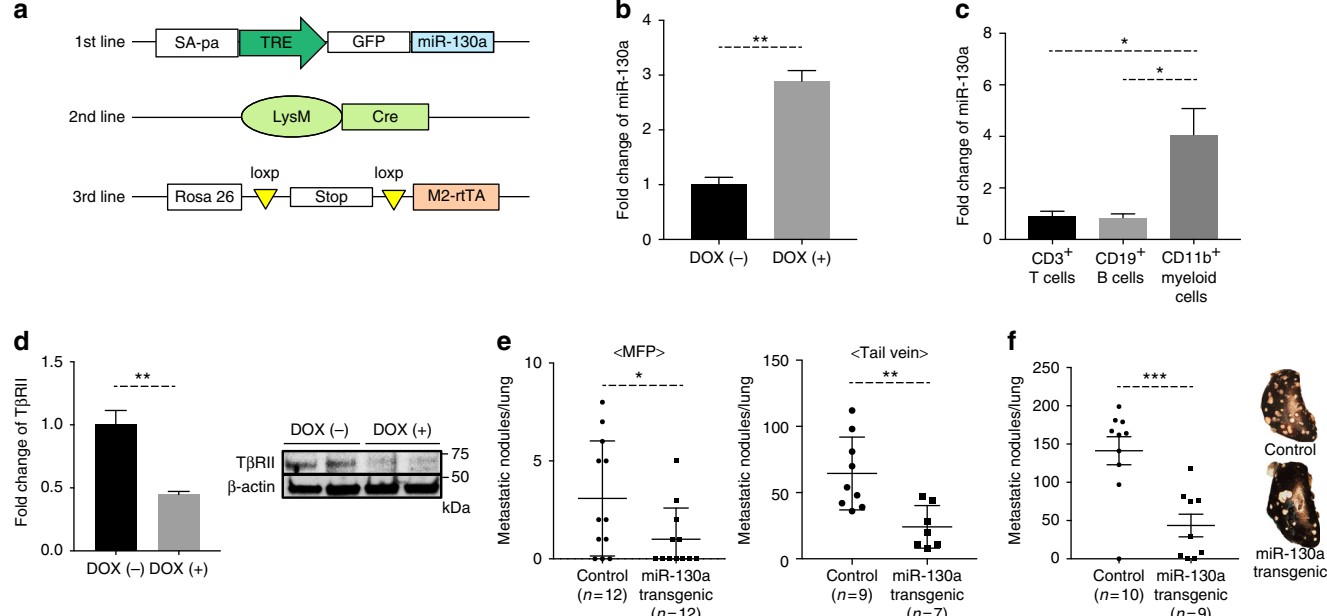

**Fig. 3** miR-130a transgenic mice showed a decreased metastasis. **a** GEM strains crossbred to produce a myeloid-specific Tet-inducible miR-130a transgenic mouse. **b** Fold change of miR-130a in Gr-1$^+$CD11b$^+$ myeloid cells from miR-130a transgenic mice treated with doxycycline (DOX), which was quantified and normalized to that of untreated mice. **c** miR-130a expression in CD3$^+$ T cells, CD19$^+$ B cells, and Gr-1$^+$CD11b$^+$ myeloid cells from the splenocytes of the miR-130a transgenic mice treated with DOX ($n = 3$). **d** Left panel: qRT-PCR of TβRII in Gr-1$^+$CD11b$^+$ myeloid cells from the spleen of E0771 tumor-bearing miR-130a transgenic mice with DOX treatment ($n = 3$); right panel: Western blot of TβRII. **e** Number of E0771 metastatic nodules in miR-130a transgenic mice treated with DOX compared with control mice (Left panel: spontaneous metastasis, $n = 12$; Right panel: experimental metastasis, $n = 7$–9). **f** Metastasis nodule counts of Lewis lung carcinoma in miR-130a transgenic mice treated with DOX compared with control mice after tail vein injection. Lung nodules were counted after 14 days ($n = 9$). Representative lungs stained by Indian ink on the right panels. The data was represented as mean±SEM, and Student's $t$ test was performed. $*p < 0.05$, $**p < 0.01$, $***p < 0.001$

many characteristics with human breast cancer, particularly its ability to spontaneously metastasize to the lungs. Balb/c mice were transplanted with HS/PCs engineered with miR-130a, miR-145, or shRNA-TβRII, without GFP reporter to avoid immune rejection in immune competent Balb/c mice. The mice were then orthotopically implanted with 4T1 cells in the MFP 10 weeks after bone marrow reconstitution. Lung metastasis was assessed 28 days later. There was a decreased TβRII expression in Gr-1$^+$CD11b$^+$ myeloid cells sorted from peripheral blood (Fig. 2d). The number of metastatic lung nodules was significantly decreased in mice that received miR-130a-, miR-145- and shRNA-TβRII-engineered HS/PCs compared to the control group (Fig. 2e), with no difference in primary tumor size (Supplementary Fig. 2i). These results demonstrate that miR-130a and miR-145 decreased TβRII expression in myeloid cells and inhibited lung metastasis.

**Decreased metastasis in miR-130a transgenic mice**. To verify that the decreased metastasis results from overexpression of miR-130a or miR-145 in myeloid cells, we produced transgenic mice in which the miRNAs were inserted in the ColA1 locus under control of the tetracycline responsive element (TRE) promoter and with a GFP reporter. One founder line was established for miR-130a. These transgenic mice were bred with R26-M2-rtTA knockin and LysM-Cre to generate triple transgenic mice with myeloid-specific, Tet-inducible miRNA expression (hereafter referred to as miR-130a transgenic) in vivo (Fig. 3a). LysM-Cre transgenic mice are well characterized and have been used in many studies to delete genes specifically in myeloid cells[35,36]. We were unable to produce the miR-145 transgenic mouse line, as embryonic stem cells (ES) expressing miR-145 failed to incorporate into the germline, perhaps due to a high level of miR-145 expressed in ES cells (Supplementary Fig. 3a). miR-130a

transgenic mice were obtained at the expected frequency and were not distinguishable from wt littermate controls. miR-130a was expressed ~3-fold higher in the transgenic myeloid cells when mice were treated with doxycycline (DOX) than that from the untreated littermate control mice (Fig. 3b). In addition, miR-130a expression did not change in CD3$^+$ T cells and CD19$^+$ B cells indicating the myeloid specificity of the transgene (Fig. 3c). To examine whether DOX-induced miR-130a expression indeed occurs in Gr-1$^+$CD11b$^+$ myeloid cells, GFP$^+$Gr-1$^+$CD11b$^+$ myeloid cells were sorted from DOX-treated miR-130a transgenic mice and cultured in vitro. miR-130a expression level was 4-fold higher in DOX-treated GFP$^+$Gr-1$^+$CD11b$^+$ myeloid cells than the untreated control cells (Supplementary Fig. 3b), similar to the in vivo induction level. As expected, the TβRII expression was significantly decreased in sorted Gr-1$^+$CD11b$^+$ myeloid cells when mice were treated with DOX (Fig. 3d). There was no difference in myeloid subsets between control and the miR-130a transgenic mice treated with DOX (Supplementary Fig. 3c). Importantly, there was a significant reduction of lung metastasis nodules in these miR-130a transgenic mice that received orthotopic MFP injection or tail vein injection of E0771 cells (spontaneous and experimental metastasis models) (Fig. 3e), with no differences in primary tumor weight (Supplementary Fig. 3d). This result could also be recapitulated in the experimental metastasis of Lewis Lung carcinoma (3LL) (Fig. 3f). Altogether, these data provide additional evidence for the critical role of myeloid miR-130a in inhibiting tumor metastasis.

**miR-130a and miR-145 reprogram Gr-1$^+$CD11b$^+$ cells and enhance immunity**. We next investigated the underlying mechanisms for the roles of miR-130a and miR-145 in metastasis reduction. We published previously that downregulation of TβRII

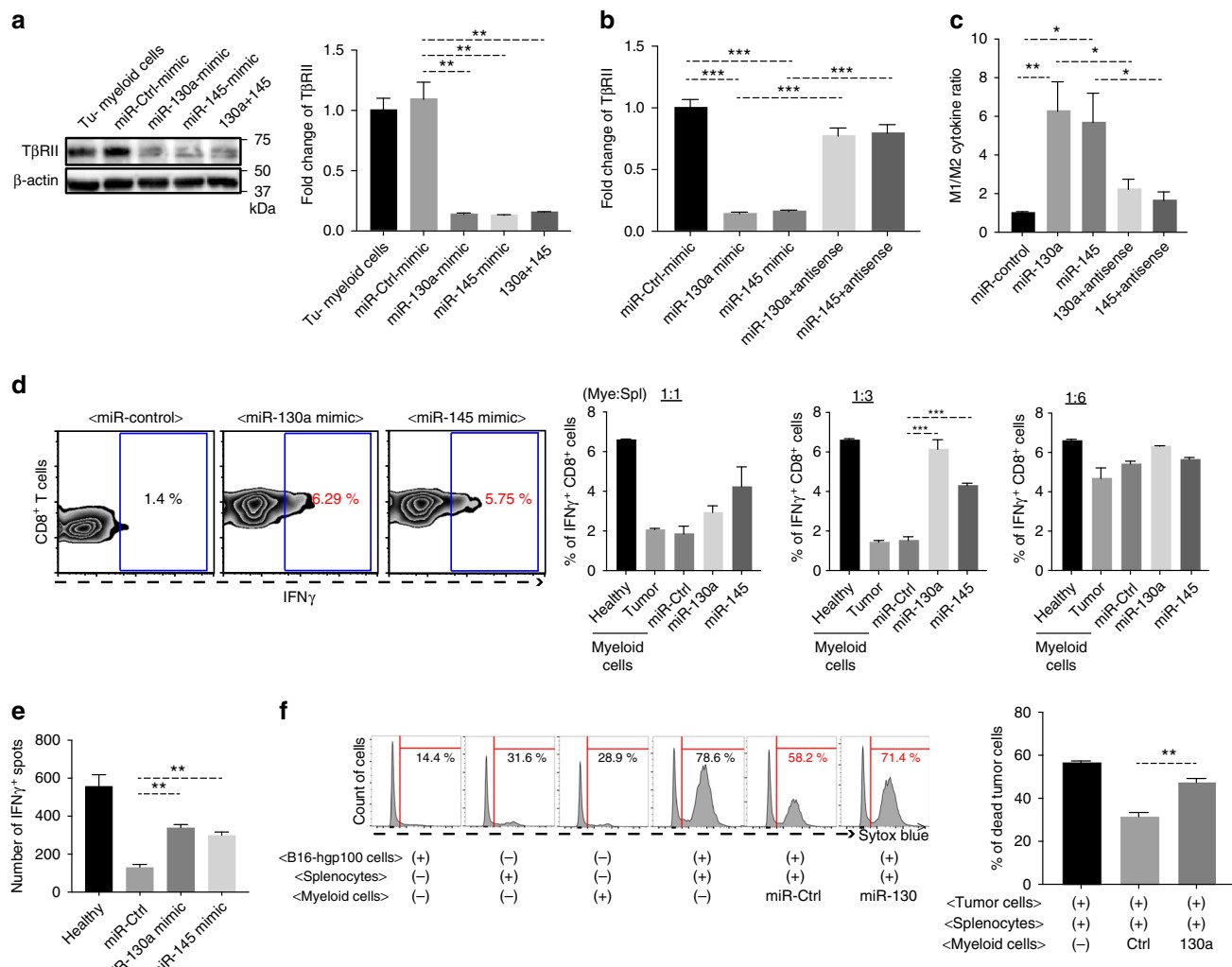

**Fig. 4** miR-130a & miR-145 reprogram Gr-1$^+$CD11b$^+$ cells and enhance host immunity. **a** TβRII Western blot (left) and qRT-PCR (right) of Gr-1$^+$CD11b$^+$ cells from 4T1 tumor-bearing mice 24 h after miR control, miR-130a and miR-145 electroporation. **b** qRT-PCR of TβRII after electroporation of miR-130a or miR-145 with or without miRNA antisense inhibitors. **c** M1/M2 cytokine ratio post treatment with miRNA electroporation with or without miR-130a or miR-145 antisense inhibitors ($n = 5$). The ratio of M1/M2 cytokines was calculated by dividing each M1 cytokine (TNFα, IL-12, GM-CSF) to M2 cytokine (IL-10, TGFβ1, IL-4) as described in Material and Methods. **d** Left panels: flow cytometry of IFNγ$^+$CD8$^+$ T cells from splenocytes of TCR-HA transgenic mice co-cultured with Gr-1$^+$CD11b$^+$ myeloid cells with or without miRNA, which were pulsed with HA peptide. Right panels: quantitative data. The Gr-1$^+$CD11b$^+$ cells (Mye) to splenocytes (Spl) ratio is indicated for each panel. **e** IFNγELISPOT of splenocytes from TCR-HA transgenic mice co-cultured with Gr-1$^+$CD11b$^+$ cells following miRNA electroporation. **f** CTL assays: flow cytometry of B16 melanoma tumor cells (Sytox blue positive are dead) expressing hgp100 co-cultured with splenocytes from Pmel-1 transgenic mice and Gr-1$^+$CD11b$^+$ cells from spleens of miR-130a transgenic, quantitative data on right. The percentage of targeted dead tumor cells = total dead cells + subtraction of dead cells from a single culture of tumor cells, myeloid cells, and splenocytes. All data are presented as mean±SEM, and Student's *t* test was performed. *$p < 0.05$, **$p < 0.01$, ***$p < 0.001$

in Gr-1$^+$CD11b$^+$ cells decreases M2 cytokines, resulting in an elevated ratio of M1/M2 cytokines[10]. We thus measured the expression levels of both M1 cytokines (TNF-α, IL-12, GM-CSF) and M2 cytokines (IL-4, IL-10, TGFβ1) from Gr-1$^+$CD11b$^+$ cells with or without ectopic expression of miR-130a and miR-145. The ratio of M1/M2 cytokines was calculated by dividing each M1 cytokine by M2 cytokine as previously published[37]. The culture supernatant of Gr-1$^+$CD11b$^+$ cells from mice with their HS/PCs transduced with miR-130a, miR-145, and shRNA-TβRII showed an increased ratio of M1/M2 cytokines (Supplementary Fig. 4a). In addition, Gr-1$^+$CD11b$^+$ cells electroporated with miR-130a or miR-145 mimics showed a decrease in TβRII expression in both protein and mRNA levels compared to the control (Fig. 4a). The combination of miR-130a with miR-145 did not show any additive effect (Fig. 4a). Further verification with antisense probes of miR-

130a or miR-145 revealed a diminished effect of the miRNAs on TβRII down-regulation (Fig. 4b). As expected, the ratio of M1/M2 cytokines was significantly higher with miR-130a or miR-145 electroporation, and antisense probes of miR-130a or miR-145 reversed this effect (Fig. 4c; Supplementary Fig. 4b). Importantly, myeloid cells with ectopic expression of miR-130a or miR-145 increased IFNγ-producing CD8$^+$ T cells in a co-culture, in which the splenocytes were isolated from TCR-HA transgenic mice, which TCR is specific for hemagglutinin (HA) (Fig. 4d). This result was clearly shown at the 1:3 myeloid cell: splenocyte ratio, which was further confirmed by an ELISPOT assay (Fig. 4e). In an antigen-specific T cell cytotoxicity assay in which the B16 melanoma tumor cell expressing hgp100 were co-cultured with splenocytes from Pmel-1 transgenic mice, Gr-1$^+$CD11b$^+$ cells from the spleen of miR-130a transgenic mice enhanced T cell cytotoxicity compared

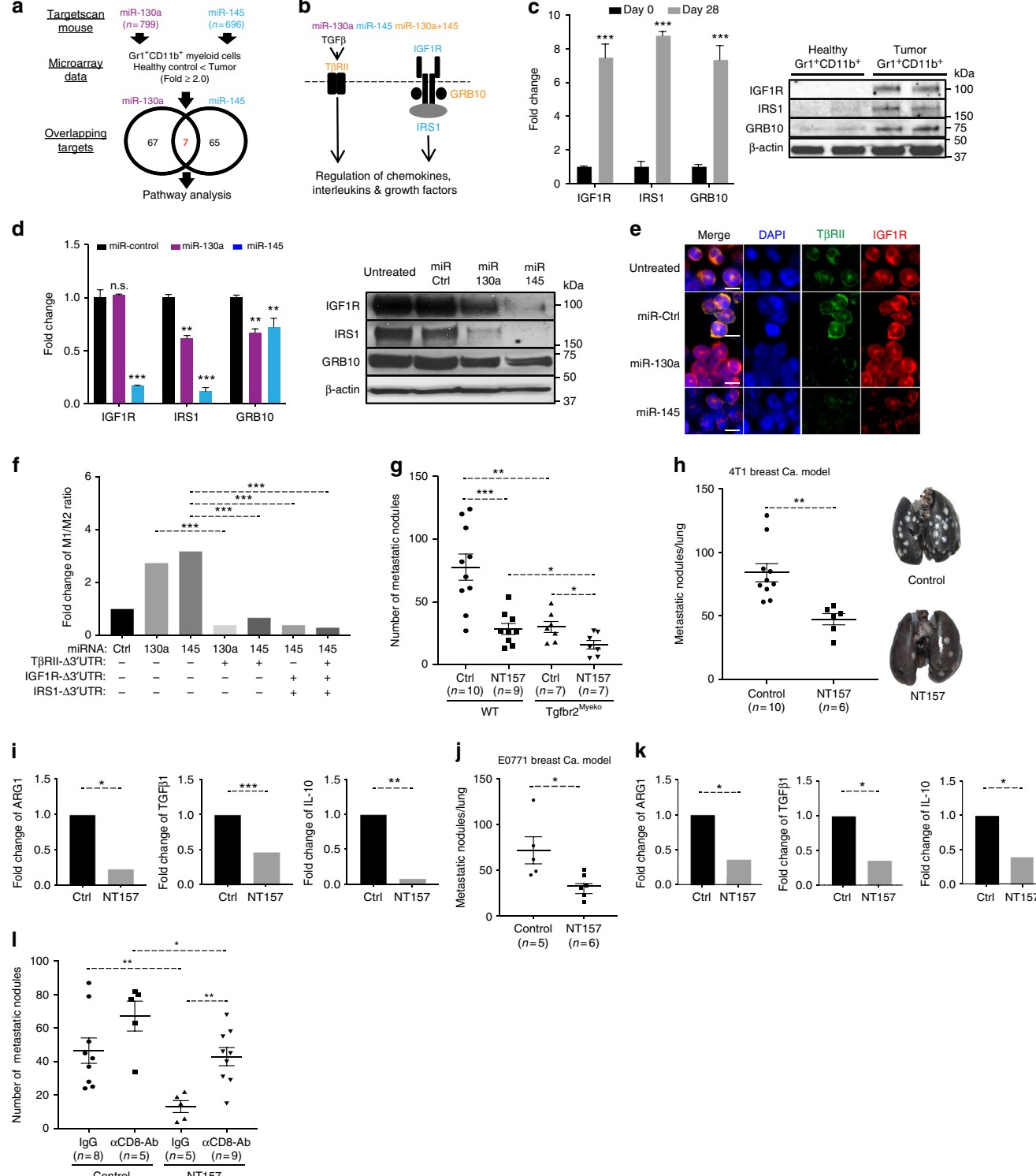

those from control mice (Fig. 4f; Supplementary Fig. 4c). Altogether, our results suggest that miR-130a or miR-145 reprogramed the myeloid cells into a M1 function and enhance host anti-tumor immunity.

**miR-130a and miR-145 targeted gene networks.** miRNAs can target multiple RNAs based on partial sequence complementarity, and induce mRNA degradation, or inhibition of protein translation[18,19,38]. We next investigated the gene networks targeted by

miR-130a and miR-145 using TargetScan Mouse software. We identified 799 and 696 genes potentially targeted by miR-130a and miR-145, respectively (Fig. 5a). These genes were then intersected with the differential RNA expression (fold > 2.0) comparing Gr-1+CD11b+ cells from tumor-bearing mice with healthy control (Fig. 5a; Supplementary Data1). There were 67 genes targeted by miR-130a and 65 targeted by miR-145, with 7 genes targeted by both miR-130a and miR-145 (Fig. 5a; Supplementary Table 2). Ingenuity Pathway Analysis (IPA) further identified multiple mediators of IGF1R signaling pathways as miR-130a and/or

miR-145 targets, including insulin-like growth factor 1 receptor (IGF1R), growth factor receptor-bound protein (GRB10), and insulin receptor substrate 1 (IRS1), a downstream mediator that is increased upon activation of IGF1R signaling[39] (Fig. 5b). Indeed, the mRNA and protein expression of IGF1R, IRS1, and GRB10 was increased in Gr-1$^+$CD11b$^+$ cells sorted from spleens of 4T1 tumor-bearing mice compared with that from healthy control mice (Fig. 5c). Electroporation of miR-130a or miR-145 mimics in these myeloid cells decreased both mRNA and protein expression of IGF1R, IRS1, and GRB10 (Fig. 5d, e). To investigate the roles of TGFβ and IGF signaling pathways in miR-130a and miR-145 regulations of myeloid immune function, overexpression constructs of *Tgfbr2*, *Igf1r*, and *Irs1* without 3′-UTR were utilized to prevent the mRNA degradation of TβRII, IGF1R and IRS1 in Gr-1$^+$CD11b$^+$ cells by miR-130a or miR-145. Myeloid cells with TβRIIΔ3′-UTR, IGF1RΔ3′-UTR, as well as IRS1Δ3′-UTR reversed the increase in M1 and M2 cytokine ratio by miR-130a and miR-145 (Fig. 5f; Supplementary Fig. 5a). Our data suggest that in addition to the TGFβ signaling pathway, IGF1R signaling is another key target of miR-130a and miR-145. Interestingly, NT157 decreased phosphorylation of IGF1R, as well as the expression of TβRII protein and mRNA in Gr-1$^+$CD11b$^+$ cells (Supplementary Fig. 5b, c), indicating a crosstalk of TGFβ and IGF1R signaling pathways in myeloid cells. Consistently, when 4T1 tumor-bearing mice with myeloid TβRII deficiency (TβRII$^{MyeKO}$) or wildtype were treated with NT157, an inhibitor of IGF1R signaling, there was a synergistic anti-metastasis effect compared with that from TβRII$^{MyeKO}$ or NT157 treatment alone (Fig. 5g). This effect was not due to decreased TβRII as TβRII in myeloid cells was absent in these mice (Supplementary Fig. 5c). However, this tumor phenotype could come from effects on tumor cells or the host immune compartment.

While the effect of IGF1R signaling on tumor cells is well established, the effect on the immune response of a tumor-bearing host is unknown. Thus, in the current studies, the effect of IGF1R signaling pathway on immunosuppression of Gr-1$^+$CD11b$^+$ cells and metastasis phenotype were further investigated in 4T1 tumor-bearing mice treated with NT157. NT157 was given 10 days after tumor inoculation and administrated three times per week (70 mg/kg through i.p.) for 3 weeks. There was a decreased primary tumor weight (Supplementary Fig. 5d; left), and importantly, a decreased number of metastasis lung nodules (Fig. 5h). The direct effect of NT157 on metastasis was carried out using mice with similar primary tumor weight to discount the possible effect of primary tumor size on metastasis readout (Supplementary Fig. 5d; right). NT157 did not affect the frequency of Gr-1$^+$CD11b$^+$ myeloid cells, or the myeloid subsets CD11b$^+$Ly6C$^+$ or CD11b$^+$Ly6G$^+$ (Supplementary Fig. 5e, f). Interestingly, Gr-1$^+$CD11b$^+$ cells from treated 4T1 tumor-bearing mice showed a decreased expression of ARG1, TGFβ1, and IL-10 (Fig. 5i), factors that are critical in mediating immune suppression in Gr-1$^+$CD11b$^+$ cells. In E0771 experimental metastasis model in which the mice received tail vein injection of E0771 cells ($1 \times 10^5$) on day 12 after mammary fat pat injection (Schematic supplementary Fig. 5g), NT157 decreased tumor weight (Supplementary Fig. 5h; left), and lung metastasis (Fig. 5j) in mice with similar primary tumor weight (Supplementary Fig. 5h; right), the latter indicating a direct effect of NT157 on metastasis. NT157 treatment also decreased expression of ARG1, TGFβ1, and IL-10 in sorted Gr-1$^+$CD11b$^+$ cells from these mice received E0771 cell injection (Fig. 5k). Further, CD8$^+$ T cell depletion in 4T1 tumor-bearing mice treated with NT157 diminished some of the inhibitory effect of NT157 on metastasis (Fig. 5l; Supplementary Fig. 5i, 5j) and the primary tumor size (Supplementary Fig. 5j; left). However, this metastasis reduction effect did not reach the level of CD8 depletion alone, indicating a direct tumor effect of NT157 in addition to host immune response. Taken together, these data demonstrate that miR-130a and miR-145 target IGF1R signaling pathways, and IGF1R inhibitor NT157 alleviates immune suppression and inhibits metastasis.

**Correlation with human cancers.** The correlation of patient survival with miR-130a and miR-145 expression in breast cancer tissues was analyzed using METABRICS dataset from Kaplan–Meier Plotter, which showed that lower levels of miR-145 or both 130a and 145 were significantly correlated with a decreased survival compared with high expression level groups (Fig. 6a). These results were also observed in liver cancer tissues (Supplementary Fig. 6a). However, the miR-130a and miR-145 expression in tumor tissues could come from tumor cells or tumor-infiltrating myeloid cells. We next co-cultured peripheral blood immature myeloid cells (CD33$^+$CD34$^+$CD15$^+$) from healthy blood donors with the human breast cancer cell line MDA-MB231. miR-130a and miR-145 were decreased in these co-cultured myeloid cells compared to myeloid cells cultured alone (Fig. 6b). Of interest, the expression levels of miR-130a and miR-145 were lower in both monocytic and granulocytic subsets from cancer patients with advanced gastrointestinal (GI) cancer than those from healthy donors (Fig. 6c; Supplementary Fig. 6b).

We further investigated the correlation of miRNA-targeted genes with human tumor progression. We have previously reported that increased myeloid TβRII level correlates with clinical stage of lung cancer, with stage III/IV showing higher TβRII level than those from stage I/II[10]. Here we focused on the correlation of myeloid IGF1R, IRS1, and GRB10 with human tumor progression. In breast cancer and pancreatic cancer cohorts (GSE27567 and GSE49641 respectively), increased GRB10 level was found in peripheral blood myeloid cells from breast and pancreatic cancer patients compared with those from

---

**Fig. 5** Gene networks targeted by miR-130a & miR-145. **a** Identification of miRNA targeted genes from TargetScan mouse database, which was intersected with mRNA expression microarray data comparing tumor Gr-1$^+$CD11b$^+$ cells with those from healthy control mice. Seven targets were common for miR-130a and -145. **b** IPA analysis of gene networks targeted by miR-130a (purple), miR-145 (blue), or both (orange) involving TGFβ and IGF pathways. **c** Validation of the major pathway mediators comparing tumor-associated myeloid cells with those from healthy controls, qRT-PCR (left) and Western blot (right). **d** qRT-PCR (left) and Western blot (right) from Gr-1$^+$CD11b$^+$ cells ex vivo treated with miR-130a or -145 or control mimics. **e** Immunofluorescence images of TβRII (Green), IGF1R (red), and DAPI (blue) in Gr-1$^+$CD11b$^+$ cells from the spleen of 4T1 tumor-bearing mice. scale bar: 10 μm. **f** M1/M2 cytokine ratio post restorations of TβRII, IGF1R, and IRS1 in Gr-1$^+$CD11b$^+$ cells that overexpress miR-130a or miR-145. The ratio of M1/M2 cytokines was calculated by dividing each M1 cytokine (TNFα, IL-12, GM-CSF) to M2 cytokine (IL-10, IL-4) as described in Material and Methods. **g–l** metastasis reduction by IGF1R inhibitor NT157: **g** The number of metastatic nodules of 4T1 tumor-bearing Tgfbr2$^{MyeKO}$ and WT mice treated with NT157 ($n = 7$–10). **h** The number of metastatic nodules in 4T1 tumor-bearing mice that received IGF1R inhibitor NT157. **i** qRT-PCR of ARG1, TGFβ1, and IL-10 in Gr-1$^+$CD11b$^+$ cells from 4T1 tumor-bearing mice treated with NT157 ($n = 5$). **j** The number of metastatic nodules from mice bearing E0771 tumor treated with NT157 ($n = 5$–6). **k** qRT-PCR of ARG1, TGFβ1, and IL-10 in Gr-1$^+$CD11b$^+$ cells from E0771 tumor-bearing mice treated with NT157 ($n = 4$). **l** The number of metastatic nodules of 4T1 tumor-bearing mice treated with NT157 with CD8α neutralizing antibody or IgG as control ($n = 10$). Mice with similar tumor weight were used for metastatic nodule counts (Supplementary Fig. 5j). Data are presented as mean±SEM, and Student's *t* test was performed. *$p < 0.05$, **$p < 0.01$, ***$p < 0.001$

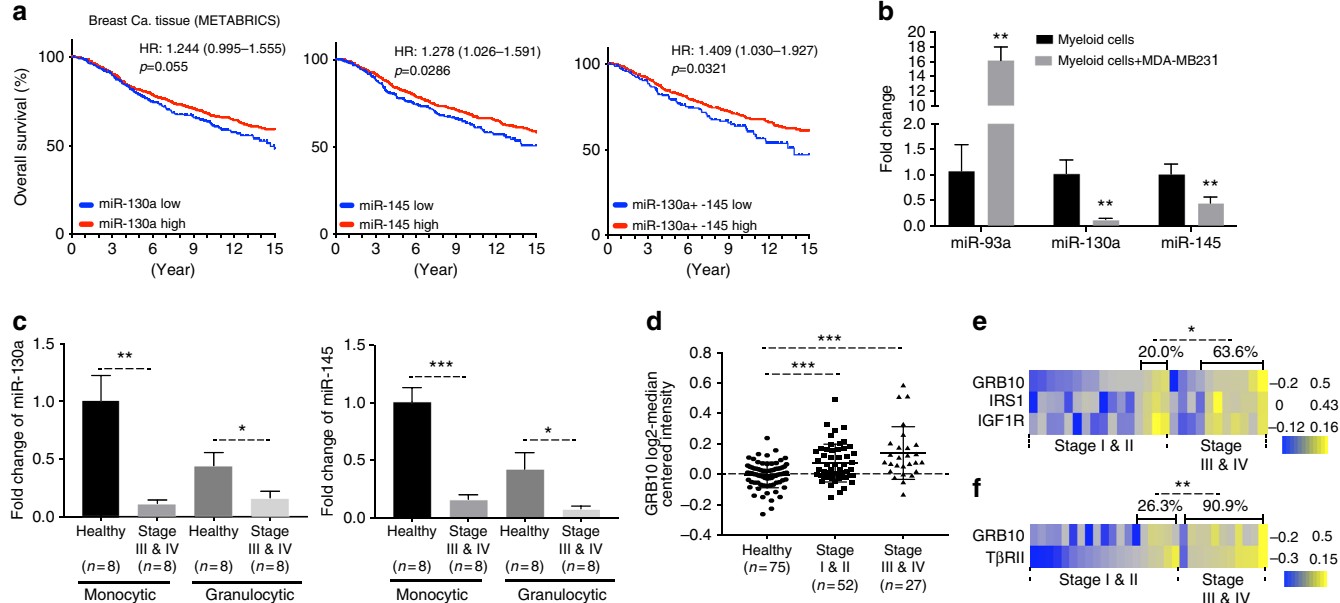

**Fig. 6** miR-130a and miR-145 targeted genes and correlation with human cancer. **a** Kaplan–Meier survival curve (METABRICS data) for breast cancer patients with high & low levels of miR-130a (left), miR-145 (middle), and both miR-130a and 145 (right) in tumor tissues. Log-rank test was used to analyze the survival data. HR hazard ratio with its 95% confidence interval. **b** Fold change of miR-130a and miR-145 expression in human immature myeloid cells co-cultured with human breast cancer cell line MDA-MB231, miR-93 as a positive control. **c** qRT-PCR for fold changes of miR-130a (left) and miR-145 (right) in sorted human monocytic and granulocytic subsets from peripheral blood of healthy donors ($n = 8$) or patients with advanced GI cancer ($n = 8$). **d** Correlation of GRB10 levels in human peripheral blood mononuclear cells (PBMCs) (GSE20189) with human lung cancer stages. **e** Heat map showing a correlation of three key IGF pathway mediators (IGF1R, GRB10, and IRS1) with stages of human lung cancer (GSE20189). **f** Correlation of GRB10 and TβRII expression levels (key mediators of both IGF and TGFβ pathways) with stages of human lung cancers (GSE20189). Yellow and blue colors indicate increased and decreased expression, respectively. The data were plotted with patients divided into stage I/II vs III /IV groups. The percentage of patients with higher expression level was calculated for each group as indicated. The data are represented as mean ± SEM, and Student's $t$ test or $\chi^2$-square test in (**e**, **f**) was performed *$p < 0.05$, **$p < 0.01$, ***$p < 0.001$

healthy control donors (Supplementary Fig. 6c, d). In addition, GRB10 levels in peripheral blood myeloid cells correlated with stages of human lung cancer (GSE20189), with stage III/IV lung cancer showing higher levels of GRB10 than stage I/II (Fig. 6d). Further, more lung cancer patients with higher levels of all three molecules (GRB10, IRS1, and IGF1R) were in stage III/IV group than the stage I/II group (Fig. 6e). Of great interest, patients with higher levels of both GRB10 and TβRII were in the more advanced disease group than those with lower levels of both GRB10 and TβRII (Fig. 6f). Together, these data suggest the relevance of miR-130a & miR-145 and the targeted TGFβ and IGF signaling pathways in human cancer progression.

**Therapy with miRNA mimics.** Restoration of tumor suppressive miRNAs in tumor cells or targeting the oncomiRs in the tumor microenvironment has shown therapeutic potential for cancer treatment[32,33,40]. We next explored whether systemic delivery of synthetic miRNA could reduce metastasis. This strategy was hypothesized to target the myeloid compartment because (1) a decrease of miR-130a and miR-145 in myeloid cells of tumor-bearing mice; (2) the higher phagocytic capacity of myeloid cells compared with lymphoid lineages. miR-130a and miR-145 mimics purified by high-performance liquid chromatography were used for preclinical mouse studies. First, the kinetics of miRNA mimics were examined through qRT-PCR on plasma and Gr-1⁺CD11b⁺ cells from peripheral blood, after delivering through tail vein injection. miR-130a was at its highest level in plasma and Gr-1⁺CD11b⁺ cells 5–6 hours (h) after delivery. Plasma miR-130a and miR-145 levels declined rapidly and reached a minimum level about 24 h after injection. miR-130a

and miR-145 in Gr-1⁺CD11b⁺ cells were evidently higher than that in the serum and were retained at 1.5–2.5 fold over baseline even after 72 h (Fig. 7a; left). TβRII expression was decreased in Gr-1⁺CD11b⁺ cells 24 h after miRNA mimics injection (Fig. 7a; right). As expected, miR-130a and miR-145 levels were significantly higher in Gr-1⁺CD11b⁺ cells compared to CD3⁺ T lymphocytes or CD19⁺ B cells at 24 h post miRNA mimics injection (Fig. 7b). In therapeutic mouse models, miRNA mimics were injected twice weekly at 1 mg/kg starting 10 days after tumor injection for a total of 4 weeks. In addition, we explored combination therapy with Paclitaxel (PAC) (schematic Supplementary Fig. 7a), a chemotherapeutic agent widely used in breast cancer treatment, with a dose of 6 mg/kg i.v. to minimize host toxicity yet augment the miRNA mimics effect. No additive reduction with the combined therapy of PAC and miR-130a or miR-145 compared to the PAC, miR-130a, or miR-145 treatment alone was observed (Fig. 7c). There was no difference in primary tumor weight (Supplementary Fig. 7b). The combination of miR-130a with miR-145 did not show an additive effect in reducing metastasis or primary tumor weight (Fig. 7c; Supplementary Fig. 7b). However, there was a clear metastasis reduction in mice treated with miR-130a or miR-145 alone compared to the control miRNA (Fig. 7c), an expected result based on reduced metastasis in miR-130a or miR-145 HS/PCs transduction (Fig. 2e), and miR-130a transgenic mice (Fig. 3e, f).

Mechanistically, ex vivo co-culture assays showed that Gr-1⁺CD11b⁺ cells from mice treated with miRNA mimics increased the number of IFNγ⁺CD8⁺ T cells compared with those from control mice (Fig. 7d). In situ hybridization with miRNA mimic probes showed a significant enrichment of miRNA mimics in CD11b⁺ myeloid cells compared with the E-cadherin⁺ tumor

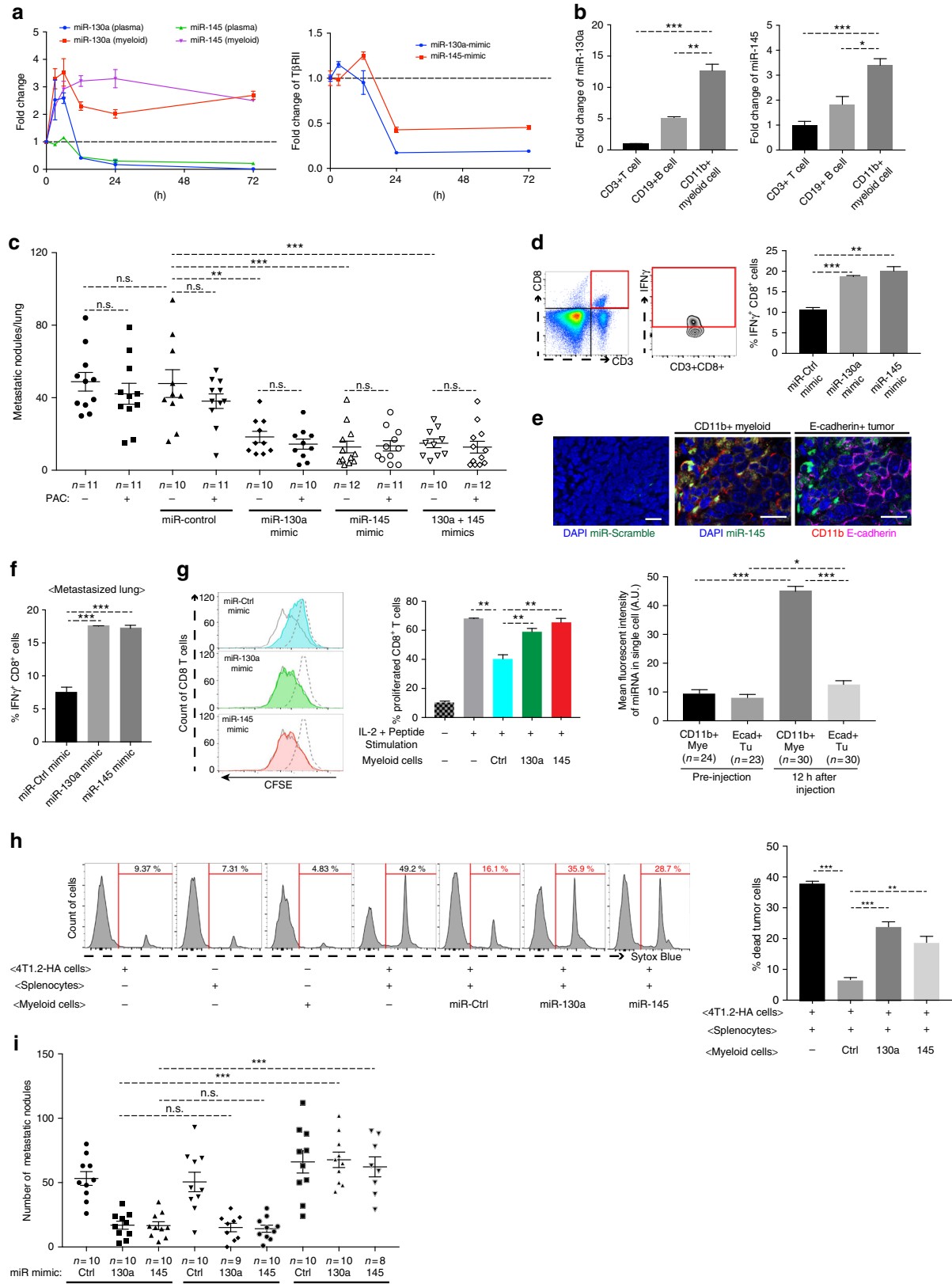

cells in metastatic lungs of mice that received miRNA mimics injection (Fig. 7e). Consistently, there was an increase in IFNγ+ CD8+ T cells in the metastatic lungs of those mice (Fig. 7f). Of notice, miRNA mimics treatment did not change the myeloid subsets (Supplementary Fig. 7c). Importantly, the proliferation of

antigen-specific CD8 T cells was increased when co-cultured with Gr-1+CD11b+ cells from these treated mice (Fig. 7g; Supplementary Fig. 7d). In addition, CD8 T cell cytotoxicity was also increased when 4T1.2 tumor cells expressing HA were co-cultured with CD8+ T cells from TCR-HA transgenic mice and

Gr-1[+]CD11b[+] cells from mice treated with miRNA mimics (Fig. 7h). CD8 but not CD4 T cells depletion in mice treated with miRNA mimics diminished the inhibitory effect of miRNA mimics on metastasis (Fig. 7i: Supplementary Fig. 7e) and primary tumor weight (Supplementary Fig. 7f). These data together with the miRNA engineered bone marrow transplantation suggest miR-130a and miR-145 could decrease tumor metastasis, improve host immune environment, thus providing novel therapeutic options in metastasis treatment.

## Discussion

Tumor-associated myeloid cells promote distant organ metastasis[1]. One important underlying mechanism is through the production of soluble factors, that not only skew a pro-tumor microenvironment at primary and distant sites but also diminish host immunity systemically[3,4,7,8]. Published studies from our group and others demonstrate that increased TβRII expression, and elevated TGFβ signaling pathways, are important in the immune suppressive function of Gr-1[+]CD11b[+] cells, and in progression and metastasis of breast cancer[10,13,14], skin cancer[15], colon cancer[16], and glioma[17]. Here, we report that down-regulation of miR-130a and miR-145 are responsible for the increased expression of TβRII in these immature myeloid cells. Ectopic expression of miR-130a and miR-145 reprogrammed Gr-1[+]CD11b[+] myeloid cells resulting in improved host anti-tumor immunity and decreased metastasis. In addition to TβRII, miR-130a and miR-145 also target multiple mediators in the IGF1R signaling pathway that correlated with tumor stage of cancer patients. Therapeutically, miR-130a and miR-145 mimics inhibited metastasis and enhanced anti-tumor immunity. Our studies identify a novel option to alter the cytokine milieu and metastatic microenvironment for therapeutic intervention of cancer metastasis.

One of the challenges in cancer biology is to skew type 2 polarization to type 1 to promote anti-tumor immunity. Current approaches have had limited success, including Toll-like receptor activation alone, or in combination with co-stimulatory signals, as well as intra-tumoral delivery of IL-2, IL-12, TNFα, IFNγ, and soluble receptors that capture immunosuppressive cytokines[41]. Neutralization of TGFβ, IL-4, IL-10 or IL-35, and inhibition of STAT3 activity, also showed limited efficacy[41,42]. Immune checkpoint blockade against CTLA-4 and members of the PD-1/PD-L1 pathway are encouraging in both the pre-clinical and clinical settings[43], however, acquired resistance has been observed[44,45]. TGFβ signaling pathway is important in the polarization of immune cells in the tumor microenvironment[46], yet there is no known approach that effectively exploits TGFβ pathway to reverse type 2 to type 1 polarization. Our data demonstrate the miR-130a and miR-145 could be utilized to target myeloid TGFβ pathway that is critical in the Th2 polarization of myeloid cells.

miR-130a and miR-145 were found to directly target TβRII expression in Gr-1[+]CD11b[+] cells in a TβRII-3′-UTR luciferase reporter assay, and other in vitro & in vivo assays (Figs. 1f, 2c, d, 3d, 4a, b). Indeed, TβRII is regulated by miRNAs in several cell types including pluripotent stem cells[47], mesenchymal stem cells[48,49], neonatal leukocytes in allergic rhinitis[50], hepatic stellate cells[51], and cancer cells[52]. While there is a scarcity of literature regarding miR-130a in tumor progression, down-regulation of miR-130a was found to be critical in M1 to M2 polarization and correlated with poor prognosis and increased lung cancer metastasis[53]. In addition, miR-130a is important for down-regulation of Smad4 in granulocytic precursors and regulation of the timed expression of C/EBP-ε during granulopoiesis[54,55]. For miR-145, a tumor suppressive function has been reported[56–58]. In particular, loss of miR-145 is critical in myelodysplastic syndrome[59]. These findings agree with the pro-immunity and anti-tumor functions observed in our studies. We want to point out that the utilization of shRNA-TβRII in parallel with miR-130a and miR-145 independently confirm that a decreased TβRII diminished the metastasis phenotype (Fig. 2d, e).

In addition to TβRII, miR-130a and miR-145 also target multiple mediators in IGF1R signaling. Recently, IGF1R signaling was found to be important in the pathogenesis of myeloproliferative neoplasms[60] and M2-like macrophage activation[61]. In addition, IGF-1 produced by macrophages is important in microvesicle-dependent communication between macrophages and epithelial cells[62], as well as resistance to IGF1R inhibition in glioma treatment[63]. Interestingly, the crosstalk of TGFβ with IGF signaling pathways has been reported[64–66]. Perhaps, targeting TGFβ signaling may induce compensatory activation of IGF pathways, the miR-130a, and miR-145 approaches target the TGFβ-IGF associated molecular network, should be helpful to overcome this problem. This insight raises question and awareness to the reductionism that often occurs when using protein-coding gene signatures and single protein pathway markers for complex phenotypes in cancer etiology.

In our effort to test whether miR-130a and miR-145 could be utilized for therapy, we performed systemic delivery of synthetic in vivo ready miRNAs mimics with the intention to exploit the phagocytic property of the myeloid cells to deliver them into the

**Fig. 7** Reduced metastasis and enhanced anti-tumor immunity by miRNA mimics. **a** qRT-PCR of miR-130a and −145 in the plasma and Gr-1[+]CD11b[+] cells (left), as well as TβRII in Gr-1[+]CD11b[+] cells (right) from 4T1 tumor-bearing mice ($n = 6$/each time point) at indicated times after systemic delivery of miRNA mimics through TVI. **b** qRT-PCR of miR-130a (left) or miR-145 (right) in Gr-1[+]CD11b[+] myeloid cells, CD3[+] lymphocytes, and CD19[+] B cells sorted from peripheral blood of mice ($n = 3$) 24 h after injection of miRNA mimics. **c** Metastasis nodule counts from mice that received miR control, miR-130a, and miR-145 mimics as a single agent or in combination with Paclitaxel (PAC) ($n = 13$). **d** Flow cytometry of IFNγ[+]CD8[+] cells from spleens of TCR-HA transgenic mice co-cultured (ex vivo) with Gr-1[+]CD11b[+] cells from 4T1 tumor-bearing mice that were treated with miR-130a or miR-145 mimics. **e** Upper panels: miRNA in situ hybridization: miR-145 mimic distribution (green) in metastatic lungs of 4T1 tumor-bearing mice 12 h after miR mimic injection, with DAPI (blue), CD11b (red), and E-cadherin (Magenta). Scale bar: 20 μm. Lower panel: the quantitative data of miRNA mean fluorescent intensity in myeloid cells or tumor cells in metastatic lung of mice with miRNA mimic injections ($n = 23$–30). **f** Flow cytometry of IFNγ[+]CD8[+] T cells at metastasized lung from mice treated with miR-130a or miR-145 mimics ($n = 3$). **g** CD8 T cell proliferation assay: flow cytometry of CSFE-labeled CD8[+] T cells from splenocytes of TCR-HA transgenic mice co-cultured with sorted Gr-1[+]CD11b[+] myeloid cells from the spleen of mice treated with miR-130a or miR-145 mimic injection (left), with quantitative data (right). **h** CTL assays: flow cytometry of 4T1.2 tumor cells (Sytox blue positive are dead) expressing HA co-cultured with splenocytes from TCR-HA transgenic mice and Gr-1[+]CD11b[+] cells from the spleen of mice treated with miR-130a or miR-145, quantitative data (right). The percentage of targeted dead tumor cells = total dead cells minus dead cells from single culture of tumor cells, myeloid cells, and splenocytes. **i** Number of metastasis in 4T1 tumor-bearing mice treated with miR mimics with CD4 or CD8α neutralizing antibody or IgG control. The data are presented as mean ± SEM, and Student's $t$ test was performed. *$p < 0.05$, **$p < 0.01$, ***$p < 0.001$

myeloid compartment. We were successful in this strategy as miR-130a and miR-145 were specifically delivered to the myeloid cells, but much less so to T or B cells (Fig. 7a, b). In addition, while the miRNAs in serum declined rapidly, miR-130a and miR-145 in myeloid cells remained at 1.2–3-fold higher even after 72 h (Fig. 7a). The sustained miRNA levels in myeloid cells are likely important in metastasis reduction in tumor-bearing mice. These results support therapeutic strategies that target the tumor microenvironment thus overcome the physiological and cellular barriers to delivery into the tumor cells[40]. In the field, miRNA mimics and miRNA antagonists are two of the main approaches to miRNA-based cancer therapies that restore the expression of tumor-suppressive miRNAs or inhibit oncomirs[33]. In fact, miR-34 replacement may be among the first miRNA mimics to reach the clinic as it is downregulated in numerous cancers and inhibits malignant growth by repressing genes involved in various oncogenic signaling pathways[67]. Intravenous delivery of miR-141, -190, and miR-219 in vivo inhibits osteoclast activity and reduces osteolytic bone metastasis[28]. In addition, cell-based, biomolecule based and polymer-based delivery of downregulated miRNAs (tumor suppressors) in cancer has shown success[68]. Restoration of miRNAs in tumor cells showed therapeutic potential for cancer treatment[28,31–33]. Unlike those studies with a focus on tumor cells, we focused on myeloid-specific delivery with the intention to reprogram the tumor microenvironment. Our data support a unique application and efficacy of miRNA mimics in targeted delivery into myeloid cells.

In summary, our studies find that downregulation of miR-130a and miR-145 is responsible for increased TβRII in myeloid cells of tumor-bearing mice and in human myeloid cells co-cultured with tumor cells. Restoration through ectopic expression, miRNA mimics or use of miR-130a transgenic mice, significantly decreased tumor metastasis. The underlying mechanisms are mediated through the reprogramming of the myeloid cells and improved host anti-tumor immunity. In addition to TβRII, miR-130a, and miR-145 also target the IGF1R signaling network. We demonstrate that miR-130a and miR-145 targeting myeloid TGFβ signaling or IGF1R inhibitor could be exploited to decrease tumor metastasis through improved anti-tumor immunity. We anticipate that restoration of miR-130a and miR-145 in Gr-1$^+$CD11b$^+$ cells could alter the cytokine milieu and metastatic microenvironment thus reduce the metastases burden in cancer patients.

## Methods

**Mice and cell lines**. BALB/c and C57Bl/6 mice (female, 6–8-week-old) from Charles River were used to perform the in vivo tumor studies. miR-130a transgenic mice were generated at Transgenic Mouse Model (TMM) Laboratory, NCI Mouse Repository (http://mouserepository-staging.ncifcrf.gov/escell/availableCells.asp), from a single ES clone in which the miR-130a was inserted into ColA1 locus under control of the TRE promoter . These mice were then bred with Rosa-LSL-rtTa [B6. Cg-*Gt(ROSA)26Sor*$^{tm1(rtTA*M2)Jae}$/J] and LysM-Cre [B6.129P2-*Lyz2*$^{tm1(cre)Ifo}$/J] mice from The Jackson Laboratory to generate myeloid-specific, tet-inducible miR-130a. One founder line was established for miR-130a. For miR-145, ES cells failed to go germline likely due to a very high level of miR-145 expression. The Pmel-1 [B6.Cg-/Cy Tg(TcraTcrb) 8Rest/J] and TCR-HA [CBy.Cg-*Thy1*$^a$Tg(TcraCl4, TcrbCl4)1Shrm/ShrmJ] transgenic mice specific to human glycoprotein 100 (hgp100) 25–33 peptide and influenza hemagglutinin (HA) 518–526 peptide (IYSTVASSL) were obtained from the Jackson Laboratory and the Cancer Inflammation Program at National Cancer Institute (NCI). The mice with myeloid-specific *Tgfbr2* deletion (Tgfbr2$^{MyeKO}$) were established as described previously[10]. All animal protocols were approved by National Cancer Institute's Animal Care and Use Committee. Murine 4T1 and E0771 cell lines were used for in vivo tumor studies. Alternatively, Lewis lung carcinoma (3LL) cell line was used for experimental metastasis in miR-130a transgenic mice. Human breast tumor cell line MDA-MB-231 was also used in co-culture assay with Gr-1$^+$CD11b$^+$ cells. HEK293 cells was used for lentiviral production. These cell lines were purchased from the American Type Culture Collection. B16 melanoma cells expressing hgp100, and 4T1.2 breast cancer cells expressing HA, are generous gifts from Surgery Branch and Dr. Thomas Sayers in Cancer Inflammation Program, CCR,

NCI. All cell lines were confirmed to be mycoplasma negative by MycoAlert$^{TM}$ Mycoplasma Detection Kit (Lonza). Doxycycline (1 μg/ml: Sigma-Aldrich) and doxycycline diets (625 mg/kg or 1.6–2.7 mg daily: Harlan Teklad) were used for myeloid-specific miR-130a induction in miR-130a transgenic mice.

**Spontaneous and experimental metastasis**. For orthotopic metastasis, the 3–5 × 10$^5$ mammary tumor 4T1 or E0771 cells were injected into the #2 MFP. For experimental metastasis, LLC (1 × 10$^5$) or E0771 (1 × 10$^5$) were injected into the tail vein. The number of lung metastases were evaluated after 4–5 weeks by Indian ink staining. Sample size were chosen based on previous data for metastasis assays.

**In vivo systemic delivery of miRNA mimics and chemotherapy**. Tumor-bearing mice were treated with in vivo ready miR-130a-3p, miR-145-5p or -control mimics (1 mg/kg, Thermo Fisher Scientific) by tail vein injection twice a week with or without paclitaxel (6 mg/kg, Teva Generics, NDC # 0703-4764-01), which was given via tail vein starting day 10 after tumor injection. The number of lung metastasis, tumor weight, and spleen weight were evaluated 4 weeks later. Gr-1$^+$CD11b$^+$ myeloid cells were isolated from miRNA mimic-treated mice and used for immunological assays.

**Targeting IGF1R signaling in 4T1 or E0771 tumor models**. NT157 (HY-100037 Medchemexpress LLC), a selective inhibitor targeting the IGF1R/IRS signaling pathway, was used for in vivo pre-clinical studies. NT157 was dissolved in 20% 2-hydroxypropyl-β-cyclodextrin (2-HP-β-CD: H-107 Sigma) at 5 mg/ml. NT157 (70 mg/kg) or vehicle was injected three times per week through intraperitoneal route. For mouse tumor models, 4T1 or E0771 cells (3 × 10$^5$) were injected into #2 MFP of female Balb/c or C57Bl/6 mice respectively. For 4T1 orthotopic mammary tumor model, the treatment was given starting on day 10 after tumor inoculation. For the E0771 metastasis model, E0771 cells were injected in the MFP and followed by 1 × 10$^5$ cells through the tail vein on day 12. After NT157 treatment for 3 weeks, tumor weight, spleen weight, and the number of lung metastases were assessed. Gr-1$^+$CD11b$^+$ myeloid cells were isolated from spleens of treated mice and used for various assays.

**Bone marrow transplantation**. Mouse bone marrow was flushed from both femurs and tibias, and was enriched for HS/PCs using immunomagnetic column (Stem Cell Technologies)[69]. Enriched 10$^6$/ml HS/PCs were stimulated overnight with a cocktail of cytokines containing (10 ng/ml M-SCF, 10 ng/ml TPO, 10 ng/ml FLT3-L, 10 ng/ml IL-6) and transduced with 10$^8$ Transduction Units/ml lentivirus in the presence of ViralPlus Transduction Enhancer (Applied Biological Materials Inc). Cells were washed 24 h after transduction and 10$^6$/100 μl cells were injected intravenously into lethally irradiated mice (900 rads). Some of the transduced cells were cultured with 40 ng/ml GM-CSF and 40 ng/ml IL-6 or 4T1 tumor culture supernatant for 4 days to differentiate into immature myeloid cells, and the genomic integration was determined by conventional PCR and FACS analysis for GFP. Transplanted mice were used for in vivo experiments after 10-week reconstitution.

**Flow cytometry, cell sorting, and IFNγ-ELISPOT**. Single cell suspensions were made from spleens or peripheral blood of healthy control or 4T1 tumor-bearing mice, as well as TCR-HA or miR-130a transgenic mice, or mice that received bone marrow transplants. Cells were labeled with fluorescence-conjugated antibodies: CD3 (145-2c11, BD Biosciences 1:100), CD4 (GK1.5, BD Biosciences 1:100), CD8 (53-6.7, BD Biosciences 1:100), CD19 (1D3, BD Biosciences 1:100), Gr-1 (RB6-8C5, 1:100), CD11b (M1/70, BD Biosciences 1:100), Ly6C (AL-21, BD Biosciences 1:200), Ly6G (1A8, BD Biosciences 1:100), IFNγ (XMG1.2, BD Biosciences 1:100), CD117 (2B8, BD Biosciences 1:100), ScaI (E13-161.7, Biolegend 1:100), HLA-DR (TU36, BD Biosciences 1:100), CD14 (MφP9, BD Biosciences 1:100), CD15 (W6D3, BD Biosciences 1:100), Sytox blue (S34857, Thermo Fisher Scientific 1:2000), CFSE (C34554, Thermo Fisher Scientific 1:100) and 7-AAD (00-6993-50, BD Biosciences 1:1000) and analyzed on a FACS Calibur, Canto II or LSR II flow cytometer (BD Biosciences). For sorting, CD3$^+$CD4$^+$ or CD3$^+$CD8$^+$ T cells, CD3$^-$CD19$^+$ B cells, and Gr-1$^+$CD11b$^+$, CD11b$^+$Ly6C$^+$ or CD11b$^+$Ly6G$^+$ myeloid cells, or GFP$^+$ or GFP$^-$Gr-1$^+$CD11b$^+$ myeloid cells were sorted by the FACSAria flow cytometer (BD Biosciences) or magnetically activated cell sorting (MACS) with CD11b and Gr-1 microbeads per the manufacturer's protocol (Miltenyi Biotec). For human immature myeloid cells, peripheral blood from healthy donors was obtained from NIH blood bank. The cells were labeled with anti-CD33-PE (WM53, BD Biosciences 1:100), anti-CD15-PE (W6D3, BD Biosciences 1:100), and anti-CD34-PE (581, BD Biosciences 1:100) and sorted by MACS per the manufacturer's protocol (Miltenyi Biotec). For human monocytic and granulocytic subsets, peripheral blood samples were obtained from healthy donors (n = 8) and patients with advanced GI cancer (n = 8) at the GI Malignancies Section, Medical Oncology Branch at NCI. Written consent was obtained from all patients before blood sampling on a research protocol approved by the NCI Institutional Review Board (NCI-11-c-0112). Cells were enriched by anti-CD14, anti-CD15, and anti-HLA-DR microbeads (Miltenyi Biotec) as recommended[70]. For IFNγ intracellular staining of CD8$^+$ T cells, splenocytes from TCR-HA transgenic mice were stimulated with leukocyte activation cocktail with golgi plug (BD Pharmingen) for 4

h, then pulsed with HA 518-526 peptide (1 μg/ml, AS-21158 ANASPEC), which then followed by co-culture with different ratios of myeloid cells overexpressing miR-130a, miR-145, or miRNA control. For IFNγ intracellular staining of CD8[+] T cells for metastasized lung of mice treated with miR mimic injections, single cell suspension was obtained after tissue dissociation with 1 mg/ml Collagenase, 120 μg/ml Dispase, and 200 μg/ml DNase, the cells were then fixed and permeabilized using a Fixation/Permeabilization kit (BD Biosciences) and stained for IFNγ. FlowJo software (Tree Star, Ashland, OR, USA) was used for analysis of flow cytometry data. For IFNγ-ELISPOT, the assays were conducted per manufacturer instruction (BD Bioscience) and spot numbers were automatically counted by ELISPOT counter (Immuno Capture 6.3.5 software).

**Mouse M1/M2 cytokine assay by Bio-plex and ELISA.** Mouse cytokine Th1/Th2 kit from BioRAD (Bio-Plex Pro™ Mouse Cytokine Th1/Th2 Assay #M60-00003J7) and mouse TGF-β1 DuoSet ELISA kit from R&D systems (DY-1679-05) was used to determine the secretory cytokines from Gr-1[+]CD11b[+] transduced with miR-130a and miR-145 lentivirus vectors or miRNA mimics. These transduced cells were cultured in serum-free medium for 24 h, and supernatants from the cells were collected. The amount of secreted cytokines and the ratio of M1/M2 was calculated by dividing each M1 cytokine (TNFα, IL-12, GM-CSF) to M2 cytokine (TGFβ1, IL-10, IL-4)[37]. Briefly, each individual cytokine was measured by Bio-plex, and each individual M1/M2 ratio (TNFα/TGFβ1, TNFα/IL-10, TNFα/IL-4, IL-12/TGFβ1, IL-12/IL-10, IL-12/IL-4, GM-CSF/TGFβ1, GM-CSF/IL-10, and GM-CSF/IL-4) was calculated, and summed up for each of miRNA and control. For preventing the mRNA degradation of TβRII, IGF1R and IRS1 in Gr-1[+]CD11b[+] cells by miR-130a or miR-145, overexpression constructs without 3′-UTR were utilized including FerH-mouse *Tgfbr2* (Protein Expression Laboratory in Frederick National Laboratory for Cancer Research), pReceiver-mouse *Igf1r* (GeneCopoeia), pBABE-mouse *Irs1* (Addgene). The M1 and M2 cytokine ratio was calculated as indicated above.

**Proliferation and CTL assay of CD8 T cells.** For antigen-specific CD8 T cell proliferation regulated by myeloid cells, total splenocytes from TCR-HA transgenic mice were labeled with CFSE dye, which were then co-cultured with Gr-1[+]CD11b[+] cells sorted from spleens of tumor-bearing mice received miR mimic injection, in culture medium contained 100 IU/ml IL-2 and 1 μg/ml HA peptide, with a 1:3 ratio (myeloid cells: labeled splenocytes) for 4 days to expand CD8 T cells. The suppressive effect of myeloid cells on CD8 T cell proliferation was evaluated by flow cytometry analysis of the CFSE-labeling intensity of CD8 T cells.

For antigen-specific cytotoxic T cell activities regulated by myeloid cells, whole splenocytes from Pmel-1 or TCR-HA transgenic mice were cultured in RPMI1640 medium containing 100 IU/ml IL-2, 1 μg/ml hgp100 (25–33) or HA (518–526) peptide, 5% fatal bovine serum, and 100 μg/ml streptomycin/penicilin for 4 days to expand antigen-specific CD8 T cells. B16 melanoma cells expressing hgp100 (1 × 10[5]) or 4T1.2 breast cancer cells expressing HA (1 × 10[5]) were co-cultured with expanded CD8[+] T cells, with the addition of Gr-1[+]CD11b[+] myeloid cells sorted from the spleen of miR-130a transgenic mice or mice treated with miR mimics at 1:1 ratio (myeloid cells: T cells) for 24 h. The suppressive effect of myeloid cells on antigen-specific CTL was evaluated by flow cytometry analysis of dead tumor cells. The percentage of dead tumor cells targeted was analyzed by flow cytometry of Sytox Blue (which labels dead cells): % dead tumor cells targeted = % all dead cells in co-culture - % spontaneous dead tumor cells - % dead myeloid cells - % dead splenocytes. The percentage of dead cells from a single culture of tumor cells, myeloid cells, and splenocytes was subtracted to get specific tumor cell killing.

**CD4[+] or CD8[+] T cell depletion.** For in vivo depletion of CD4 or CD8 T cells, neutralizing antibodies of CD4 or CD8α or IgG2b control (GK1.5 or YTS169.4 clone, or LTF-2, Bio X Cell, 20 μg/mouse) were intraperitoneally injected every 3 days starting from the day 0 of 4T1 injection until mice were sacrificed and evaluated for lung metastasis.

**Isolation of genomic DNA, total RNA, miRNA, and PCR.** For posttranscriptional stability of mRNA, Gr-1[+]CD11b[+] cells from healthy control or tumor-bearing mice were treated with 1 μM Actinomycin D (A1410, Sigma-Aldrich) for 0 to 8 h at 37 °C. RNA was extracted using RNeasy Mini Kit (Qiagen). RNA was measured using qRT-PCR (Applied Biosystems). 18sRNA was used to normalize for both GAPDH and TβRII expression.

Genomic DNA or total RNA was extracted from sorted cells or subsets using ZR-DUCT™ DNA/RNA Miniprep kit (Zymo Research) and RNeasy Mini kit (Qiagen), and cDNA was synthesized with high capacity cDNA reverse transcription kit (Applied Biosystems #4368813). Conventional PCR was performed with AmpliTaq Gold DNA Polymerase to observe genomic integration in bone marrow HSC. Uncropped scans of gels are provided in Supplementary Figure 8. qRT-PCR was performed using FastStart Universal SYBR Green Master (Rox) (04913850001 Roche) and ABI 7500 Fast real-time PCR system (Applied Biosystems). Primers are listed in supplementary Table 3.

miRNA from serum or cells was extracted using miRVana miRNA isolation kit (Ambion AM1560), which was converted to cDNA by Taqman MicroRNA Reverse Transcription kit (CA94404 Applied Biosystems) and TaqMan microRNA assays

for miR-130a-3p, -145-5p, -19a, -93-5p,, and U6 were purchased from life technologies. Relative gene expression was determined using Taqman 2 × Universal PCR Master mix (4324018 Applied Biosystems). The relative expression level of miRNAs or mRNA transcripts was normalized to that of internal control U6 or GAPDH by using the $2^{-\Delta\Delta Ct}$ cycle threshold method.

**NanoString miRNA array, sample preparation and data analysis.** RNA (100 ng) was used as input for the NanoString platform. Mature miRNAs were ligated overnight at 65 °C to a species-specific tag sequence (miR tag) followed by hybridization with a nCounter mouse miRNA Expression Assay CodeSet. The hybridized CodeSet was purified using a nCounter Prep Station and analyzed through a nCounter Digital Analyzer. The raw data were imported into a nSolver program that uses a normalization factor to account for technical noise. The normalization factor was generated using the geometric mean of the top 100 miRNAs for each sample; raw counts were multiplied by this sample specific normalization factor to produce the normalized data. In addition, six internal negative controls and six positive controls were included in the nCounter miRNA Expression assays for medium and high stringency calculation, respectively.

**Western blotting.** Protein extracts from Gr-1[+]CD11b[+] cells were analyzed by Western blot. The following primary antibodies were used: TβRII (AF532, RD 1:1000), GRB10 (07-2182, MILLIPORE 1:1000), IGF1R (9750, CST 1:1000), phospho-IGF1Rβ (Y1135/1136, and Y1150/1151) (3024, CST 1:1000), IRS1 (3407, CST 1:1000), β-actin (SC69879, Santa Cruz 1:2000). Anti-mouse/rabbit/goat secondary antibodies were purchased from Bio-Rad (1:3000 dilution). Uncropped scans of western blots are provided in Supplementary Figure 8.

**Luciferase reporter assays.** The pGL3 luciferase vector, in which the 3′-UTR of Tgfbr2 was cloned into XbaI site, was a generous gift from Dr. Tariq Rana (Stanford University). To measure luciferase activity, 200 ng of Tgfbr2 vectors and 50 ng of pRL-TK (Renilla Luciferase) were transfected into 1 × 10[5] Hela cells along with 100 nM of respective miRNA (Life Technologies). 48 h post transfection, cell lysates were collected by adding 250 μl of passive lysis buffer and placing on a rocking platform at room temperature for 15 min. 20 μl of cell lysate was used to assess the luciferase activity as per the manufacturing protocol (Dual-Luciferase® Reporter Assay System Promega).

**miRNA Constructs and lenti virus production.** All entry clones for over-expression of miR-130a, miR-145, and control were in Biosettia intronic format. miRNA constructs containing intronic expression cassettes for miRNAs were subcloned into lentiviral vectors with a CD11b promoter. All lentiviral expression clones had lentiviral inverted LTR repeats and were propagated in a low recombination *E. coli* strain STBL3 (Invitrogen) using the appropriate antibiotic and low-temperature growth (30 °C) to maintain plasmid stability.

For lentivirus production, the 2[nd] generation system, psPAX2, and pMD2G were used. Briefly, HEK293T cells (10 × 10[6]) were seeded in a 15-cm dish for approximately 24 h, followed by plasmid-lipofectamine (Thermo Fisher Scientific) in serum-free medium. 24 h later, fresh medium was added in a total volume of 15 ml containing viral boost reagent (ALSTEM). 30 h later, cell supernatants were filtered and concentrated using a high-speed centrifuge at 12,000 × g. Titers were determined using serial dilutions of concentrated virus in Raw264.7 macrophage cell line or Lenti-X GoStix (Clontech).

**Immunofluorescence.** Cells were fixed in 100 % cold methanol for 10 min and permeabilized for 10 min in 0.3% Triton X-100 in PBS. After blocking with 5% normal Donkey serum (Ab7475, Abcam) for 1 h, cells were incubated overnight at 4 °C with following primary antibodies: TβRII (AF532, RD 1:100) and IGF1R (9750, CST 1:100). The cells were then incubated with Alexa Fluor 488 and 594 donkey anti-goat/rabbit IgG (H + L) (A-11055/R-37119, Life Technologies 1:500) for 2 h, and then mounted with Antifade Mounting medium with DAPI (VEC-TASHIELD). Fluorescence was examined by microscopy (Olympus IX-81).

**miRNA in situ hybridization.** miRNA in situ hybridization was performed using the miRCURY LNA™ microRNA ISH Optimization kit contained double-DIG-labeled LNA™ detection probe for miR-145-5p and scramble-miR negative control (EXIQON). Formalin-fixed tissue sections of metastatic lungs from tumor-bearing mice were incubated with 15 μg/ml Proteinase-K for 10 min at 37 °C. The miR-145 (20 nM) or control probe (40 nM) were hybridized at 60 °C for 16 h. After washing in SSC and blocking by 5% normal donkey serum for 1 h, hybridized sections were incubated overnight at 4 °C with primary antibodies and anti-DIG conjugated with Alexa 488 (IC7520G, Novus 1:100). anti-E-cadherin (24E10, CST 1:100) and anti-CD11b (ab8878, Abcam 1:100) were used to detect tumor cells and myeloid cells. The distribution of miR-145 was examined by confocal laser scanning microscope (LSM710, Zeiss).

**Human correlative studies.** Publicly available datasets of human peripheral mononuclear cells (PBMCs) from patients with breast cancer (GSE27567), pancreatic cancer (GSE49641), and lung cancer (GSE20189) were used to investigate

the correlation of miR-130a and/or miR-145-targeted gene expression levels in PBMCs with tumor progression and tumor stages. For Kaplan-Meier survival curve in breast and liver cancer patients, METABRICS data in Kaplan-Meier Plotter were used.

**Statistical Analysis**. GraphPad Prism was used for graphs and for statistics. Unless otherwise indicated, all data were analyzed using the Student's t-test or $\chi^2$-square test and are expressed as mean ± SEM. Kaplan-Meier analysis and log-rank tests were applied for survival analysis. Differences were considered statistically significant when the p-value was <0.05.

**Data availability**. The data supporting the findings of this study are available within the article and its supplementary information files, or are available from the corresponding author upon request.

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

## Acknowledgements

The construct of 3′-UTR of TβRII luciferase reporter is a kind gift from Drs Nakashima, & Tariq Rana, Sanford-Burnham Medical Research Institute, and University of California San Diego School of Medicine. We thank Dr. Thomas Sayers in Cancer Inflammation Program, CCR, NCI to share 4T1.2 cell line overexpressing HA. We thank Dr. Yanli Pang for critical reading of the manuscript. We thank all staff in FACS Core for technical assistance. We are grateful for advice from Dr. Anand Merchant in bioinformatics analysis. This work was supported by NCI intramural funding to Dr. Li Yang.

## Author contributions

H.I., S.K.V., and B.R.A. designed, planned, and performed the experiments, analyzed data and wrote a draft of the paper. S.K.V. and B.R.A. performed bone marrow transplantation experiments. J. S. provided technical assistance on animal studies. M.C.H. established miR-130 transgenic mice and was involved in animal works. H.I., B.R.A., and A.L. performed bioinformatics and statistical analysis. T.F.G. provided peripheral blood of liver cancer patients and is involved in experimental designs and manuscript revision. A. L. participated in the discussion of experimental designs, data analysis, and editing the manuscript. L.Y. initiated, organized and designed the study, supervised progress, and wrote the manuscript.

## Additional information

**Competing interests:** The authors declare no competing interests.

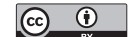

