## [Peer Review File · Nature Communications]

Reviewers' comments:

Reviewer #1 (Remarks to the Author):

Ishii et al describe the miRs 130a and 145 as potential therapeutic tools, since these miRs downregulate IGF1R signaling and TGFbR2 expression, resulting in reduced metastasis formation. These findings are innovative and potentially interesting, but at present several important shortcomings are present:

- 1) The authors should clearly indicate in the text that these CD11b+ Gr-1+ cells are isolated from the spleen. More importantly, this is an outdated way of looking at tumor-associated myeloid cells. It is mandatory to discriminate between monocytic MDSC (Ly6C-hi, Ly6G-neg, MHC-II-neg/lo) and granulocytic MDSC (Ly6C-hi, Ly6G-hi, MHC-II-neg) throughout the manuscript, as these cells behave differently.
- 2) Fig 1b: statistics are lacking
- 3) Fig 1D. It is not clear from the legend why these 7 miRs were selected in this Table, but I assume these are all miRs that target TGFbR2. 5 out of 7 of these miRs are actually upregulated in the tumor-bearing state, provoking the question why these miRs do not downregulate TGFbR2 expression. Why would miR 130a and 145 be more important?
- 4) Lentivirally transduced stem cells give rise to myeloid cells with different levels of the GFP reporter. Does this affect TGFbR2 levels? This is actually the relevant question, but is not shown by the authors.
- 5) Fig 2D. Using the GFP reporter system, only about 20% of the CD11b+ cells appear to express the cassette and, hence, the miR. I find it surprising that major effects are seen if only a rather limited subpopulation of myeloid cells expresses the miR.
- 6) GFP-positive myeloid cells were sorted. Which % of all CD11+Gr-1+ cells are GFP-positive? Is this again a minority? Are there differences between monocytic and granulocytic MDSC? TGFbR2 expression was significantly decreased in this sorted population, but again, is this true for the entire population or only a subpopulation?
- 7) The metastasis models used in C57BL/6 depend on tail vein injections, which is rather artificial. Test also a spontaneous metastasis model in the C57BL/6 strain.
- 8) The authors claim that the miRs reprogram CD11b+Gr-1+ cells, resulting in an increased tumor immunity. However, the latter point is not proven at all. They only show the in vitro capacity of transduced MDSC to stimulate IFNg production by TCR-Tg T cells. What about proliferation of these T cells? Cytotoxic activity of these T cells? These aspects are important to determine. Moreover, if the in vivo anti-metastatic effect is due to enhanced T-cell activity, the authors should be able to show increased T-cell activity at the metastatic site, the anti-tumor effect should be reversed by CD4 or CD8 T-cell depletion etc. The authors did not demonstrate any of this in vivo.
- 9) Concerning type 1/type 2 cytokine ratio. Data for individual cytokines should be shown. IFNg, IL-4 and IL-5 are typically T-cell-derived cytokines and their relevance in this context is not clear to me. It is however impossible to say whether these cytokines are produced by these myeloid cells, since only a broad type 1/type 2 ratio is given.
- 10) It is impossible to understand from the text or figure legend why TCR-HA Tg CD8+ T cells were used. Was HA protein or peptide added to the cocultures?
- 11) There was a decreased primary tumor weight when using NT157, in contrast to earlier findings with BM transfer of miR overexpressing stem cells. Explain the difference. Is the number of MDSC and their subpopulations still the same upon NT157 treatment?
- 12) NT157 downregulates TGFbR2 expression. Does this mean that IGF1R signaling is upstream of TGFbR2 expression? The relative importance of IGF1R signaling versus TGFbR2 signaling in the anti-metastatic phenotype is absolutely not clear at present and should be clarified. You should prove this by blocking IGF1R signaling in mice in which TGFbR2 expression is lowered (eg shTGFbR2 BM chimeric mice). If TGFbR2 is the most important effector molecule downstream of IGF1R, the NT157 inhibitor should not have an effect.
- 13) CD33+CD34+CD15+ cells were isolated from human peripheral blood. This combination of markers is not optimal. Again, monocytic and granulocytic cells should be looked at. Moreover, the

coculture with human cancer cell lines in vitro is too artificial. This is in my mind not a good substitute for real MDSC derived from cancer patients.

14) I don't see the relevance of the finding in the leukemia cell lines. This should be removed.

15) There is actually no proof that these miRs are implicated in the human situation as those correlative data are lacking. The involvement of IGF1R and TGFbR2 is very indirect evidence for the role of these miRs in patients.

16) The rationale of combining the miR mimics and paclitaxel is absolutely not clear. Would you expect an additive or even synergistic effect and why?

17) The in vivo treatment with miR mimetics may reprogram MDSC, but again, their effect on T cells is only tested in vitro, while no data are available about the in vivo T-cell activity.

Reviewer #2 (Remarks to the Author):

This group has previously demonstrated that Gr-1+CD11b myeloid cells from tumor bearing mice have increased expression of TGFβR11 and TGFβ signaling and that this TGFβ signaling is critical in their immune suppressive function. Thus, promoting metastasis.

In the current study they identified miR-130a and miR-145 to be downregulated in these cells and regulating and/or the cause for the upregulation of TGFβR11 and the immune suppressive functions of Gr-1+CD11B. They also identified the IGFR1 as a second target for these miRs. The presented studies are very well executed.

Nevertheless there are several conceptual concerns to be addressed to support the overall role of this axis (miR-130a, miR-145, TGFβ signaling, immune response and metastasis).

1. The studies are based on one tumor model of 4T1 breast cancer cells and I.V injections of EO771 mammary tumor cells. Is it unique to epithelial tumors or other solid tumors?

2. Injections of tumor cells into the transgenic animal of miR-130a (miR-145) were not performed (MFP of 4T1 cells).

3. The experiments with NT 157 are not novel and are not informative. The effect of NT 157 on the tumor cells themselves were not taken into account.

4. Delivery of naked miRs in vivo is hard to explain, how they are homing to the myeloid cells and not the tumor cells. The vast majority of them will not reach their target.

5. I understand that the authors would like to focus of the myeloid cells, but what about the effect of these miRs on the expression of these receptors on the tumor cells?

6. Mining the TCGA data for the expression of these miRs in various cancers and correlation with survival will strengthened the hypothesis of the current studies.

A point-by-point response to the reviewers' comments and a list of the incorporated changes are detailed below:

Reviewer #1 (Remarks to the Author):

Ishii et al describe the miRs 130a and 145 as potential therapeutic tools, since these miRs downregulate IGF1R signaling and TβRII expression, resulting in reduced metastasis formation. These findings are innovative and potentially interesting, but at present several important shortcomings are present:

1) The authors should clearly indicate in the text that these CD11b+ Gr-1+ cells are isolated from the spleen. More importantly, this is an outdated way of looking at tumor-associated myeloid cells. It is mandatory to discriminate between monocytic MDSC (Ly6C-hi, Ly6G-neg, MHC-II-neg/lo) and granulocytic MDSC (Ly6C-hi, Ly6G-hi, MHC-II-neg) throughout the manuscript, as these cells behave differently.

We have provided information on samples in the text, and added the data on MDSC subsets, monocytic (CD11b+Ly6C+) and granulocytic (CD11b+Ly6G+), as detailed below:

- 1) miR-130a & miR-145 expressions in monocytic and granulocytic subsets from 4T1 mouse model (Figure 1h) and E0771 model (Supplementary figure 1c), as well as the corresponding TβRII expression in these myeloid subsets (Fig. 1h, Supplementary figure 1c).
- 2) As shown in Supplementary figure 2g, there was no difference in monocyte and neutrophil counts comparing mice that received miRNA-engineered BMT with the vector control BMT.
- 3) There was no difference in myeloid subsets in the transgenic mice with myeloid-specific miR-130a overexpression (Supplementary figure 3c).
- 4) No changes of myeloid subsets were found in mice after IGF1R inhibitor treatment (Supplementary figure 5e).
- 5) Systemic miR mimic injections did not change myeloid subsets (Supplementary figure 7c).

In summary, miR-130a and -145 are decreased in both myeloid subsets under tumor conditions, which is consistent with an elevated TβRII expression. There was a deeper decrease in granulocytic compared to monocytic subset in the 4T1 model. This is not clear in the E0771 model. As reported by us and others, granulocytic subset is significantly expanded from 4T1 tumor-bearing mice. Thus, in the 4T1 tumor model, the miR effect is largely on granulocytic subset. In addition, miR overexpression or miR mimic injections did not change the frequency of myeloid subsets.

2) Fig 1b: statistics are lacking

We have now added statistical analysis in Figure. 1b. Thank you.

3) Fig 1D. It is not clear from the legend why these 7 miRs were selected in this Table, but I assume these are all miRs that target TβRII. 5 out of 7 of these miRs are actually upregulated in the tumor-bearing state, provoking the question why these miRs do not downregulate TβRII expression. Why would miR 130a and 145 be more important?

All the seven miRs have differences in a number of binding sites on TβRII 3'UTR. Further, not all of them are conserved throughout the species. Because TβRII is up-regulated in immature myeloid cells under tumor conditions (Cancer Discovery 2013; 3:936-951), so we choose to focus on miRs that are downregulated, which include both miR-130a and miR-145. Our data from different experimental approaches support the importance of miR-130a and miR-145 in regulating an increase in TβRII expression.

4) Lentivirally transduced stem cells give rise to myeloid cells with different levels of the GFP reporter. Does this affect TβRII levels? This is actually the relevant question, but is not shown by the authors.

Yes, TβRII levels are affected. The data is now added in Supplementary figure 2d.

5) Fig 2D. Using the GFP reporter system, only about 20% of the CD11b+ cells appear to express the cassette and, hence, the miR. I find it surprising that major effects are seen if only a rather limited subpopulation of myeloid cells expresses the miR.

6) GFP-positive myeloid cells were sorted. Which % of all CD11+Gr-1+ cells are GFP-positive? Is this again a minority? Are there differences between monocytic and granulocytic MDSC? TβRII expression was significantly decreased in this sorted population, but again, is this true for the entire population or only a subpopulation?

The initial data in Figure 2c in characterizing GFP reporter activity was from healthy conditions, which is now in Supplementary figure 2d. We now have added data for tumor condition through the *ex vivo* experiment in which the engineered HS/PCs are cultured in 4T1 tumor supernatant, which is added to Figure 2c. There were about 65% GFP+ in CD11b+ myeloid cells, and over 90% are a granulocytic subset (Figure 2b). It's worth noting that in the 4T1 metastasis model, the granulocytic subset constitutes the majority of the MDSCs, which showed GFP+ here.

We found elevated expression of miR-130a and miR-145 in the GFP+ subset compared to those in the GFP- subset (Figure 2c). Consistently, TβRII was significantly lower in miR-130a and miR-145 overexpressed myeloid cells (Figure 2c). We reason that low levels of miR in the GFP- population could result from a small population of earlier myeloid progenitor cells.

7) The metastasis models used in C57BL/6 depend on tail vein injections, which is rather artificial. Test also a spontaneous metastasis model in the C57BL/6 strain.

There are very few spontaneous metastatic mammary tumor models for the C57Bl/6 background. Experimental metastasis model is useful in dissecting later steps of metastatic process e.g extravasation and colonization. Nevertheless, per reviewer comments, we have examined E0771 spontaneous metastasis. Consistent with our earlier results, the spontaneous metastasis was inhibited in mice with myeloid specific miR-130a overexpression (Figure 3e).

8) The authors claim that the miRs reprogram CD11b+Gr-1+ cells, resulting in an increased tumor immunity. However, the latter point is not proven at all. They only show the in vitro capacity of transduced MDSC to stimulate IFN γ production by TCR-Tg T cells. What about proliferation of these T cells? Cytotoxic activity of these T cells? These aspects are important to determine. Moreover, if the in vivo anti-metastatic effect is due to enhanced T-cell activity, the authors should be able to show increased T-cell activity at the metastatic site, the anti-tumor effect should be reversed by CD4 or CD8 T-cell depletion etc. The authors did not demonstrate any of this in vivo.

We appreciate the comments and have carried out additional experiments to evaluate host anti-tumor immunity upon miR reprogramming of myeloid cells:

- 1) Increased antigen-specific CD8 T cell cytotoxicity from the co-culture with myeloid cells from miR-130a transgenic mice compared with the control myeloid cells (Figure 4f).
- 2) Increased IFN γ +CD8+ T cells in the lungs of tumor-bearing mice after systemic delivery of miR mimics (Figure 7f).
- 3) Increased CD8 T cell proliferation upon the reprogramming of immature myeloid cells in mice treated with miRNA mimics (Figure 7g).
- 4) Cytotoxicity of CD8 T cells: increased antigen-specific CD8 T cell cytotoxicity from the co-culture with myeloid cells from mice treated with miRNA mimics compared with the control myeloid cells (Figure 7h).
- 5) Lastly, *in vivo* CD8 but not CD4 depletion in mice treated with systemic miR mimics significantly diminished the decrease in metastasis (Figure 7i). These data indicate that *in vivo* anti-tumor effects of systemic miR mimic injection is due to enhanced CD8 T cell activity.

9) Concerning type 1/type 2 cytokine ratio. Data for individual cytokines should be shown. IFN γ , IL-4 and IL-5 are typically T-cell-derived cytokines and their relevance in this context is not clear to me. It is however impossible to say whether these cytokines are produced by these myeloid cells, since only a broad type 1/type 2 ratio is given.

The myeloid cells produced high levels of IL-4 (about 200 pg/ml culture supernatant) but not IL-5 or IFN γ . Per the reviewer comments, we have now used the following cytokines for evaluation: M1 (TNF α , IL-12, GM-CSF) and M2 (TGF β 1, IL-10, IL-4) in the culture supernatant from reprogrammed immature myeloid cells. The data for individual

cytokine production have been added in Supplementary figure 4a-b. The ratio of M1/M2 cytokines was increased in miR-130a or miR-145-reprogrammed myeloid cells (Figure 4c, Supplementary figure 4a right). We thank the reviewer for the comments.

10) It is impossible to understand from the text or figure legend why TCR-HA Tg CD8+ T cells were used. Was HA protein or peptide added to the cocultures?

Yes, splenocytes from TCR-HA transgenic mice were cultured in medium with IL-2 and pulsed with HA peptide for 5 days for CD8 T cell expansion. Myeloid cells from the spleen of tumor-bearing with or without miR reprogramming were added to the culture to investigate the effect on IFN γ producing CD8+ T cells. This information is in Material and Methods.

11) There was a decreased primary tumor weight when using NT157, in contrast to earlier findings with BM transfer of miR overexpressing stem cells. Explain the difference. Is the number of MDSC and their subpopulations still the same upon NT157 treatment?

BM transfer of miR-engineered HS/PCs reprogrammed myeloid cells and improved host anti-tumor immunity. The NT157 treatment could directly target tumor cells, and in addition, our data showed that it also increased anti-tumor immunity through reprogrammed immature myeloid cells (Figure 5h and 5j), which have not been previously reported. The later conclusion was supported by the *in vivo* CD8 depletion in mice treated with NT157, which diminished the effect of NT157 on tumor weight and the number of metastatic nodules (Figure 5k and Supplementary figure 5i).

NT157 treatment did not change myeloid cell subsets (Supplementary figure 5e).

12) NT157 downregulates T β RII expression. Does this mean that IGF1R signaling is upstream of T β RII expression? The relative importance of IGF1R signaling versus T β RII signaling in the anti-metastatic phenotype is absolutely not clear at present and should be clarified. You should prove this by blocking IGF1R signaling in mice in which T β RII expression is lowered (eg shT β RII BM chimeric mice). If T β RII is the most important effector molecule downstream of IGF1R, the NT157 inhibitor should not have an effect.

Yes, our data support a role of IGF1R signaling in regulating T β RII (Supplementary figure 5a and 5b). Per the reviewer comments, we performed NT157 treatment in T β RII^{MyeKO} and control mice. Our data revealed a synergistic effect in lung metastasis of mice bearing 4T1 tumors compared with that from T β RII^{MyeKO} or NT157 treatment alone (Figure 5f). In addition, NT157 reprogrammed myeloid cells and IFN γ +CD8+ T cells are critical in metastasis reduction in mice treated with NT157. These results support a crosstalk of IGF1R and T β RII signaling pathways in host immune responses.

13) CD33+CD34+CD15+ cells were isolated from human peripheral blood. This combination of markers is not optimal. Again, monocytic and granulocytic cells should be looked at. Moreover, the coculture with human cancer cell lines in vitro is too artificial. This is in my mind not a good substitute for real MDSC derived from cancer patients.

miR in cancer is a relatively new field. In addition, there are no published datasets on miR in peripheral blood myeloid cells from cancer patients. Further, breast cancer patients in later stages are hardly available due to early diagnosis by mammogram. The co-culture work was carried out due to these circumstances. We also made an effort to utilize publically available datasets from lung cancer patients. However, there are no reports on miR in the myeloid subsets. Only genes that are targeted by miR-130a and -145 could be investigated.

We appreciate the reviewer comments and tried our best to address the concerns. We collaborated with Dr. Tim Greten, who has expertise and knowledge in MDSCs from GI cancer patients. We sorted granulocytic and monocytic myeloid subsets from peripheral blood of patients with advanced stages (Supplementary figure 6a) using markers suggested from the publication (Nature Communications 2016;7:12150. doi: 10.1038/ncomms12150). miR-130a and -145 expression levels were decreased in both subsets from those patients (Figure 6c). These results support a human correlation of our studies.

14) I don't see the relevance of the finding in the leukemia cell lines. This should be removed.

We have removed the data.

15) There is actually no proof that these miRs are implicated in the human situation as those correlative data are lacking. The involvement of IGF1R and TβRII is very indirect evidence for the role of these miRs in patients.

Please see our response in **13)**. The direct involvement of IGF1R and TβRII is investigated in mouse models. The human studies are correlative, and many mechanistic studies have to be carried out *in vitro* or in mouse models.

16) The rationale of combining the miR mimics and paclitaxel is absolutely not clear. Would you expect an additive or even synergistic effect and why?

We expected a synergistic effect of the combination. Paclitaxel (PAC) is one of chemotherapeutic agents widely used in breast cancer treatment. It kills tumor cells, but does not address the immune suppression problem. So in our approach, we tried to use a low dose to reduce toxicity, and at the same time to use miRs mimic to improve anti-tumor immunity.

17) The in vivo treatment with miR mimetics may reprogram MDSC, but again,

their effect on T cells is only tested in vitro, while no data are available about the in vivo T-cell activity.

We carried out a number of critical experiments, please see our response in 8)

Reviewer #2 (Remarks to the Author):

This group has previously demonstrated that Gr-1+CD11b myeloid cells from tumor bearing mice have increased expression of TβRII and TGFβ signaling and that this TGFβ signaling is critical in their immune suppressive function. Thus, promoting metastasis.

In the current study they identified miR-130a and miR-145 to be downregulated in these cells and regulating and/or the cause for the upregulation of TβRII and the immune suppressive functions of Gr-1+CD11B. They also identified the IGFR1 as a second target for these miRs. The presented studies are very well executed. Nevertheless there are several conceptual concerns to be addressed to support the overall role of this axis (miR-130a, miR-145, TGFβ signaling, immune response and metastasis).

1. The studies are based on one tumor model of 4T1 breast cancer cells and I.V injections of E0771 mammary tumor cells. Is it unique to epithelial tumors or other solid tumors?

We used the 4T1 model, which is a well-established mouse model for breast cancer metastasis. In addition, a second model of E0771 was also used for experimental metastasis (original data) & spontaneous metastasis (added now, Figure 3e). For tumor phenotype and downregulation of the miRs, experiments were carried out in both models. For mechanistic studies, the 4T1 model was mostly used. Our data from mouse models and human correlative studies indicate the downregulation of miR may be a common phenomenon in epithelial cancers. However, this needs to be further investigated.

2. Injections of tumor cells into the transgenic animal of miR-130a (miR-145) were not performed (MFP of 4T1 cells).

miR-130a transgenic mice have a C57Bl/6 background, so 4T1 cells (Balb/c background) could not be used. Instead, E0771 breast cancer cell line was used in miR-130a transgenic mice (Figure 3e). As mentioned in the text, we were unable to establish the myeloid-specific miR-145 transgenic mouse line due to a too high level of miR-145 expression.

3. The experiments with NT 157 are not novel and are not informative. The effect of NT 157 on the tumor cells themselves were not taken into account.

The direct anti-tumor effect of NT157 is well-known, it's not the focus of our work. In this study, we discovered that NT157 also attenuated immune suppressive effects of immature myeloid cells and improve host anti-tumor immunity. NT157 reprogrammed myeloid cells (Figure 5h and j) and IFN γ +CD8+ T cells are critical metastasis inhibition in NT157 treated mice (Figure 5k). See Reviewer #1, **11) – 12)** also.

4. Delivery of naked miRs in vivo is hard to explain, how they are homing to the myeloid cells and not the tumor cells. The vast majority of them will not reach their target.

To evaluate the *in vivo* distribution of the miRNA mimics in immature myeloid cells and tumor cells in primary tumor tissues and metastatic nodules, we performed miRNA mimic *in situ* hybridization. There was significant miRNA mimic enrichment in CD11b+ myeloid cells but only minimum traces in tumor cells in the metastatic lungs of mice, 12-24 hr after miRNA mimics injection (Figure 7e). The enrichment in myeloid cells is likely benefitted from phagocytosis properties of these cells. Depletion of CD8 T cells diminished the anti-metastasis effect of miR-130a and -145, further suggesting that the miR effects are mostly on host immune compartment but rather than tumor cells.

5. I understand that the authors would like to focus on the myeloid cells, but what about the effect of these miRs on the expression of these receptors on the tumor cells?

There are several references regarding miR-130a or miR-145 effect on tumor cells. However, our data did not support a major role of such in our mouse models and studies presented here. We do not exclude the possible function of these miRs on tumor cells in different models.

6. Mining the TCGA data for the expression of these miRs in various cancers and correlation with survival will strengthened the hypothesis of the current studies.

We thank the reviewer for the suggestions. Analysis of database from tumor tissues of breast cancer patients (METABRICS) show a correlation of higher miR-130a and -145 levels with better survival (Figure 6a). While there are no published datasets on miR in peripheral blood myeloid cells from cancer patients, our co-culture of myeloid cells with breast cancer cells revealed a decrease in miRNA expression in the myeloid cells.

In collaboration with Dr. Tim Greten, who has expertise and knowledge in MDSCs from GI cancer patients, we sorted granulocytic and monocytic myeloid subsets from peripheral blood of patients with advanced stages (Supplementary figure 6a). miR-130a and -145 expression levels were decreased in both subsets from those patients (Figure 6c). These results support a human correlation of our studies.

Reviewers' comments:

Reviewer #1 (Remarks to the Author):

The authors successfully answered to most of my concerns, including a significant amount of additional data.

Reviewer #2 (Remarks to the Author):

1. I am still skeptical about the role of these two miRs on regulating the immune response via TGFPR and IRS1 on myeloid cells. The miRs can regulate so many genes and their delivery in vivo showed uptake also by the tumor cells. (Fig. 7e) Moreover, Figure 6a does not support the proposed working hypothesis, as no significant difference in the survival of patients or observed between the high and low expressions of these miRs.

2. The ratio of M1/M2 is not a good measure. What are the levels of each cytokine tested?

3. The authors still need to show the effect of NT157 treatment on the tumor cells themselves; including staining of CD31 as an indicator of angiogenesis.

Recommendation: Accept with major revision.

2nd revision for NCOMMS-17-16764A

Reviewer #1:

The authors successfully answered to most of my concerns, including a significant amount of additional data.

We are glad that the reviewer is satisfied.

Reviewer #2:

1. I am still skeptical about the role of these two miRs on regulating the immune response via TGFPR and IRS1 on myeloid cells. The miRs can regulate so many genes and their delivery in vivo showed uptake also by the tumor cells. (Fig. 7e) Moreover, Figure 6a does not support the proposed working hypothesis, as no significant difference in the survival of patients or observed between the high and low expressions of these miRs.

To investigate the roles of TGF β and IGF signaling pathways in miR-130a and -145 regulations of myeloid immune function, overexpression constructs of *Tgfb2*, *Igf1r*, and *Irs1* **without 3'-UTR** were utilized to prevent the mRNA degradation of T β RII, IGF1R and IRS1 in Gr-1+CD11b+ cells by miR-130a or miR-145. Myeloid cells with T β RII^{-3'UTR}, IGF1R^{-3'UTR} as well as IRS1^{-3'UTR} reversed the increase in M1 and M2 cytokine ratio by miR-130a and -145. These data have now added to **Figure 5f and Supplementary figure 5a**. These data, together with other data presented in the manuscript strongly suggest that both TGF β and IGF1R signaling pathways are important for immunosuppressive function of immature myeloid cells, and miR-130a and miR-145 can regulate immunosuppression in immature myeloid cells through down-regulation of T β RII and IGF1R pathways.

miRs indeed regulate a network of genes, consistently, our data show miR-130a and -145 target key molecules of TGF β and IGF signaling pathways in myeloid cells. For

the reviewer concern for the *in vivo* distribution of the miRNA mimics in immature myeloid cells and tumor cells, the miRNA mimic *in situ* hybridization showed a significant miRNA mimic enrichment in CD11b+ myeloid cells but only minimum traces in tumor cells in the metastatic lungs of mice (Figure 7e). We reasoned that the enrichment in myeloid cells is likely benefitted from phagocytosis properties of these cells. Depletion of CD8 T cells diminished the anti-metastasis effect of miR-130a and -145, further suggesting that the miR effects are mostly on host immune compartment but rather than tumor cells.

For human studies using METABRICS datasets, Figure 6a show that lower levels of miR-145 or both 130a and 145 were significantly correlated with a decreased survival compared with high expression level groups. We have now added a correlation of higher miR-130a and -145 levels with better survival in liver cancer dataset (**Supplementary figure 6a**).

2. The ratio of M1/M2 is not a good measure. What are the levels of each cytokine tested?

The data for individual cytokine measurements are in Supplementary figure 4a-b. The Ratio of M1/M2 is used to assess tumor-associated myeloid function such as tumor-associated macrophage.

3. The authors still need to show the effect of NT157 treatment on the tumor cells themselves; including staining of CD31 as an indicator of angiogenesis.

There are good number of reports regarding the effect of NT157 on tumor cells. We also showed a direct effect of NT157 on tumor cells (Figure 5l). This is not our focus.

Our novel findings here are the immunosuppressive functions of IGF signaling in immature myeloid cells, as shown in Figure 5f, 5i, 5k and 5l. Together with the data in TGF β mediated immune suppression (here and *Y. Pan et al. Cancer Discov. 2013;3:936-951*), our studies provide insights and options in therapeutic strategy against immune therapy relapse and resistance.

REVIEWERS' COMMENTS:

Reviewer #2 (Remarks to the Author):

The authors responded to some of my concerns and added new data. The paper is now ready to be published.

REVIEWERS' COMMENTS:

Reviewer #2 (Remarks to the Author):

The authors responded to some of my concerns and added new data. The paper is now ready to be published.

Thank you very much for your review.